# Application of UAV-SfM photogrammetry and aerial LiDAR to a disastrous flood: repeated topographic measurement of a newly formed crevasse splay of the Kinu River, central Japan

Atsuto Izumida[1], Shoichiro Uchiyama[1, 2], Toshihiko Sugai[1]

[1]Department of Natural Environmental Studies, the University of Tokyo, Kashiwa, Chiba 277-8563, Japan
[2]National Research Institute for Earth Science and Disaster Prevention, Tsukuba, Ibaraki 305-0006, Japan

*Correspondence to*: Atsuto Izumida (aizumida@s.nenv.k.u-tokyo.ac.jp)

**Abstract.** Geomorphic impacts of a disastrous crevasse splay that formed in September 2015 and its post-formation modifications were quantitatively documented by using repeated, high-definition digital surface models (DSMs) of an inhabited and cultivated floodplain of the Kinu River, central Japan. The DSMs were based on pre-flood (resolution, 2 m) and post-flood (resolution, 1 m) aerial light detection and ranging (LiDAR) data from January 2007 and September 2015, respectively, and structure-from-motion (SfM) photogrammetry data (resolution, 3.84 cm) derived from aerial photos taken by an unmanned aerial vehicle (UAV) in December 2015. After elimination of systematic errors among the DSMs and down-sampling of the SfM-derived DSM, elevation changes on the order of $10^{-1}$ m including not only topography but also growth of vegetation, vanishing of flood waters, and restoration and repair works were detected. If excluding changes other than topography, comparison of the DSMs showed that more than twice as large volume was eroded than deposited 300–500 m around the breached artificial levee where the topography was significantly affected. The results suggest that DSMs acquired by a combination of UAV-SfM and LiDAR data can be used to quantify, rapidly and in rich detail, topographic changes on floodplains caused by floods.

## 1 Introduction

Floods are becoming increasingly serious natural disasters as more and more people move into flood-prone areas along with their assets (Berz et al., 2001; Barredo, 2007; Kundzewicz et al., 2010). Floodplains have long been desirable locations for human habitation owing to their fertile soils and abundant water availability, and many of the world's great cities have been constructed on and have drastically modified the natural environment of floodplains, with the result that flooding in these areas, when it occurs, can be extremely hazardous (Wohl, 2014). When the levee of a river with a sandy bed collapses, crevasse splays, which are mainly composed of sand lobes, that is, discrete lobe- or finger-shaped mounds of sandy sediments, and crevasse channels (ephemeral tributaries through which water and sediments are transported onto the floodplain) form, resulting in abrupt, severe topographic changes on the floodplain (Allen, 1965; O'Brien and Wells, 1986; Bristow et al., 1999; Florsheim and Mount, 2002; Day et al., 2016). Present-day levee breaches along rivers under human control create landforms

quite different from and, in some cases, much larger than those created by levee breaches along natural rivers, because of houses and paved streets which control the distribution of sand lobes by acting as channels of flood flow (Nelson and Leclair, 2006) and because the construction of artificial levees raises the potential energy of the flood waters (Gebica and Sokolowski, 2001). It is important to determine the volume and area of erosion or deposition that can be expected to occur and how the

topographic changes are likely to be distributed on a floodplain during high-magnitude flood events, because these factors directly determine the magnitude and extent of the damages that can be expected to result from a disastrous flood event.

Recently, high-resolution topographic data acquired by innovative remote sensing methods such as aerial and terrestrial light detection and ranging (LiDAR) and structure-from-motion (SfM) photogrammetry has been widely used to evaluate the magnitude of natural hazards (Tarolli, 2014; Gomez and Purdie, 2016). In particular, the combination of SfM with aerial

photographs acquired by unmanned aerial vehicles (UAVs) can be very useful in hazardous areas subject to natural disasters such as earthquakes, volcanic eruptions, and landslides, because of the low-cost and mobile operation of UAVs (see Gomez and Purdie, 2016, for a review of each application type).

However, few quantitative descriptions of topography specifically related to a flood disaster on a floodplain are based on high-resolution topographic data obtained by using such methods. Wierzbicki et al. (2013) used post-flood LiDAR data to capture

a newly formed crevasse splay of the Vistula River, Poland, but they did not have any pre-flood data for a topographic comparison. The UAV-SfM method has basically not been used to examine floodplain topography in the context of a natural disaster, although Tamminga et al. (2015a) used it in a fluvial setting, applying it to the characterization of the morphology of a gravelly river. Although several studies have performed hydraulic analyses such as mapping floodwaters in urban areas (Feng et al., 2015) and estimation of a basin-scale sediment budget following a large flood (Croke et al., 2013), floodplain topography

has attracted less attention from researchers. Reasons may include the low relief and urbanized or agricultural land uses of floodplains (Ninfo et al., 2016), and the low preservation potential of flood-related topography in urban areas (Nelson and Leclair, 2006). Moreover, in many cases, reliable topographic data *before* a flood event which enables the investigation of topographic changes is not available, which prevents detailed quantification of flood-caused topographic changes (Tamminga et al., 2015b).

In this study, these difficulties were overcome and UAV and SfM photogrammetry were applied to the case of a disastrous flood that affected an inhabited and cultivated floodplain along the Kinu River, central Japan, in 2015. Three digital surface models (DSMs) of the research area characterized by the formation of crevasse splay before, 3 days after, and 3 months after the flood were generated by SfM photogrammetry or aerial LiDAR, and the features and post-formation modification of the topography caused by a levee breach were quantitatively documented from the perspective of both natural and artificial changes.

In addition, volume calculation was conducted using these DSMs to investigate the balance between deposition and erosion processes acted in the research area during the flood and the volumetric extent of the post-flood restoration works against the breached artificial levee and intensively eroded areas.

## 2 Study area

The Kinu River, which is 177 km long with a catchment area of 1760 km$^2$, is one of several large rivers in the Kanto Region, central Japan; it originates in the mountains north of the Kanto Region and joins the Tone River in the central Kanto Plain (Fig. 1). Volcanic Neogene and Quaternary age rocks are complexly distributed in the mountains that occupy the upper 60% of its catchment area. The remaining about 35% of the catchment is on the Kanto Plain and consists mainly of the present floodplain of the river and Pleistocene fluvial terraces. Mean annual precipitation is 1600–2100 mm in the mountains, and 1300–1500 mm on the plain.

The area affected by the 2015 flood is located about 50 km northeast from Tokyo (Fig. 1) and about 20 km upstream of the Kinu River's confluence with the Tone River. The Kinu River in this area has low sinuosity and a sandy bed with a gradient of about 1/2500, and it flows southward. The Kinu River and the Kokai River, another tributary of the Tone River, flow along the western and eastern margin, respectively, of a floodplain, and these rivers and their abandoned channels are surrounded by alluvial ridges 1–2 m higher than flood basin, which occupies the center of the floodplain. The floodplain itself is 4–8 km wide, and it is bordered on both the east and west by fluvial terraces (Fig. 1). The largest alluvial ridges, which are 1.5–2 km wide, are those to the east of the Kinu River in Joso City. The alluvial ridges in this floodplain are densely inhabited and most villages are located on them. Other part of the alluvial ridges are used as agricultural lands, predominantly paddy fields. On the other hand, irrigation canals are entirely equipped in the flood basin and abandoned channels, and almost all area are used as paddy fields.

The research area covered by the DSMs was on the southern part of an alluvial ridge along the Kinu River, the width of which was about 1.5 km (Fig. 1). Within the research area, the elevation of the ridge was high along the channel, especially near the breached levee, although the ground level there might have been raised for construction. The artificial levee was about 2 m higher than the top of the alluvial ridge. The alluvial ridge was divided by a shallow valley oriented north–south along the eastern margin of the study area, which Sadakata (1971) suggested was a past crevasse channel. In the present, an agricultural canal ran along the valley, and a network of smaller canals covered the research area. The part of the alluvial ridge in the study area was mainly used for cultivating agricultural crops; most of the area was not covered by pavement or buildings except close to the levee.

In fact, repeated flooding of the Kinu and Kokai rivers around the study area during the past several hundred years is documented in the historical record, although how each flood affected the topography of the floodplain is unclear. Two floods occurred recently along the Kokai River, in 1981 and 1986. During the 1981 flood, a crevasse splay was formed at a point 4 km upstream of its confluence with the Tone River, where the Kokai River intersected an abandoned channel, creating a sandy mound more than 60 cm thick and a crevasse channel (breach scouring) with a length of 200 m and a depth of 2 m (Iseya et al., 1982).

## 3 The 2015 flood of the Kinu River

A levee breach of the Kinu River at Joso City, Ibaraki Prefecture, which occurred on 10 September 2015, was caused by a record heavy rain during 7–11 September, particularly during 9–10 September, driven by Typhoon Etau and the extratropical cyclone it became. The total rainfall during the period of heavy rain exceeded 600 mm along the upper reaches of the Kinu

River, and 200–300 mm along the central to lower reaches (Japan Meteorological Agency , 2015). Statistical analysis by Yoshimura et al. (2016) suggests that the return time of the cumulative rainfall for a single day, two days and three days over the drainage area of the Kinu River during the 2015 flood is 95, 138 and 237 years, respectively. At a gauge station located 10 km upstream from the breached levee, the peak water level, which was more than 6 m higher than the ordinary water level, was recorded on 10 September. Kinugawa-Mitsukaido gauge station, which is at 10 km downstream of the breached point of

the Kinu River marked a peak discharge of c.a. 4000 $m^3 s^{-1}$, which is the maximum observed in history of 90 years, and a similar discharge (c.a. 3900 $m^3 s^{-1}$) occurred only once at 1949 there (Kanto Regional Development Bureau, Ministry of Land, Infrastructure, Transport and Tourism, 2015).

The levee breach and other outflows from the Kinu River occurred at 10 September inundated an area of 40 $km^2$ on the floodplain between the Kinu and Kokai rivers; this area is equivalent to one-third of the area of Joso City. The flood water

reached a depth of more than 2.5 m in the flood basin and the inundation depth of the alluvial ridges was generally 1 m or less near the breached levee, while the depth increased to 1.5 m or more in the south of the floodplain and decreased to 0.5 m or less, or none far from the breached levee (Nagumo et al., 2016). It took 10 days of pumping to remove the flood water from the levee-protected floodplain; there were two deaths, 44 injured, and 6000 evacuees of the 65,000 inhabitants of Joso City, and the flood damaged, destroyed, or inundated 5000 buildings, in addition to causing severe interruptions of public utilities

and the transportation system (Joso City, 2016). The breached levee had been temporarily repaired by two weeks after the flood, thus preventing additional flooding of the Kinu River (Kanto Regional Development Bureau, Ministry of Land, Infrastructure, Transport and Tourism , 2015). The economic loss by the flood and rainfall have been estimated to be 159.2 billion yens in Ibaraki Prefecture and 294.1 billion yens for the entire disaster by the heavy rain in September 2015 (Land, Infrastructure and Transportation Ministry of Japan, 2017). For further information of 2015 flood of the Kinu River, see

Nagumo et al. (2016) which have provided a detailed report of damages and social effects in the flooded areas.

## 4 Method

### 4.1 Data acquisition and DSM generation

#### 4.1.1 LiDAR data

Two sets of LiDAR measurement data, acquired on 15 January 2007, before the flood, and on 13 September 2015, 3 days after

the flood, were used in this study. Land use in the study area is mainly agricultural (cultivated land), except for the area next

to the levee, and it was largely bare of vegetation in winter when the first data set was acquired. Moreover, when the second data set was acquired immediately after the flood, the crops had been almost flattened by the flood waters. Thus, the data are fairly accurate without any adjustment for surface objects such as vegetation and buildings. These data sets included georeferenced orthophotos.

LiDAR data before the flood were provided by the Kanto Regional Development Bureau, Ministry of Land, Infrastructure, Transport and Tourism. The measurement point density was 1.2 points m$^{-1}$ in the east–west direction, and 1.3 points m$^{-1}$ in the north–south direction. The pixel size of the DSM was set to 2 m so that each pixel contained two measurement points on average. We generated a DSM from these data by using the 3D Analyst extension of ESRI ArcGIS 10.2.2 Desktop software. The Create TIN tool was first used to construct a triangular irregular network (TIN) of the data, and then the TIN was converted
to a DSM by using a TIN to Raster tool.

LiDAR data were acquired soon after the flood by Aero Asahi Corp. (AAC). The measurement point density was 0.43 points m$^{-1}$. This LiDAR data was converted to raster and a DSM was generated by AAC by the TIN method, and the pixel size was set to 1 m. When the LiDAR data in September 2015 was acquired, however, some areas (mostly erosional zones) were inundated by the flood water, which means that elevation data cloud not be obtained there. Thus, the elevation of the inundated
areas was interpolated from the elevations along the margins of those areas on the DSM.

During the LiDAR measurement, the position of the aircraft was determined by GNSS equipment, and the inclination of the aircraft (roll, pitch, and yaw) was logged by an internal measurement unit. The on-board position information was corrected by using ground reference stations with known coordinates, established by a static GNSS survey.

### 4.1.2 UAV–SfM data

On 26 December 2015, we conducted a flight campaign with a UAV (DJI F550 six-rotor multicopter equipped with an APM 2.6 flight controller) over the alluvial ridge where significant topographic changes had been caused by the flood to obtain high-resolution measurements and to characterize the post-formation modifications of the crevasse splay. The flight was automatically piloted by an APM Mission Planner ground station; the UAV's height was maintained at 150 m above ground level and its speed was maintained 10 m s$^{-1}$ while the photographs were being acquired. The UAV was equipped with a Ricoh
GR digital camera (focal length, 18.3 mm; resolution, 16.2 megapixels) and weighed about 2.4 kg altogether. Six flights, each 15 min long, were conducted over two hours on the morning of 26 December and 597 photos were taken at 2-seconds intervals with an 80% overlap both across and along the UAV paths. The camera was flexibly held by a picavet (see Inoue et al., 2014) so that off-nadir angle of each photo was a little different among 10–15 degrees backward.

We used Agisoft Photoscan Pro 1.2.1 software for automatic camera calibration and the SfM analysis. The DSM was
georeferenced by 15 ground control points (GCPs) obtained by a real-time kinematic (RTK) GNSS using a Trimble Geo 7X instrument and Zephyr Model 2 antenna with a horizontal and vertical accuracy of 5 cm (Fig. 3). Flat places with low relief such as a paved road were selected for the GCP locations so that local relief and objects would not affect the elevation of the

pixels around the GCPs. The ground resolution of the resulting DSM was 3.84 cm, and it covered an area of about 0.793 km$^2$. The root mean square error (RMSE) of the DSM with 15 GCPs was 2.37 cm in the east-west direction, 2.14 cm in the north-west direction, and 1.58 cm in the elevation, and 3.56 cm overall (Table 1). The reprojection error was 0.455 pixel, which was no larger than that of similar SfM analysis with images taken by UAV (Pineux et al., 2016).

The doming effect is a fundamental problem about DSM generation by SfM analysis associated with near-parallel image sets and inaccurate correction of radial lens distortion (James and Robson, 2014). The image acquisition method used in this study included some technics that mitigated the doming of the SfM-derived DSM: a high overlap rate of the images, precisely and widely placed GCPs and varying off-nadir angles. The error of each GCP between the GNSS measurement and the derived model was less than 5 cm, or the accuracy of the GNSS measurement, except for one located at the edge of the DSM. In
addition, a check point was obtained by the same GNSS measurement showed an error of 0.56 cm against the SfM-derived DSM (Fig. 2). Although the result of the camera calibration was not quantitatively tested, these numbers suggest that the doming effect of the SfM-derived DSM was sufficiently small to compare with other LiDAR DSMs.4.2 Evaluation of the horizontal accuracy of DSMs

To validate that an object was represented at the same location on all DSMs, each of which had been georeferenced
independently, the gradient of each pixel was calculated for the three DSMs, and the edges of the same houses and other buildings were identified independently on the three slope rasters (Fig. 4). In Fig. 4, high slope areas are white, so the boundaries between vertical structures such as buildings and walls or trees and the ground appear as white belts. The lines indicating building edges and walls on the December 2015 gradient map are located in the center of the high slope areas of the DSMs for the other two dates. Because the high slope areas are wide when the resolution of the original DSM is low, it is the
resolution of the DSM, rather than its horizontal accuracy, that matters when evaluating topographic changes by raster comparison.

## 4.3 Calculation of differential rasters

Raster subtraction of DSMs for successive dates allows the topographic changes occurring between the two dates to be evaluated. Comparison of the DSMs for the first two dates (before and soon after the flood) shows the direct impact of the
flood on the topography of the floodplain, and comparison of the DSMs for the second two dates shows post-formation modification of the crevasse splay.

First, the resolution of the SfM-derived DSM was resampled to 1 m to avoid local scale effects that might be derived from trivial relief of the high-resolution surface model that was not represented in the DSMs with lower resolution (Fig. 5). The resolution and distribution of the pixels of the down-sampled DSM were set to the same as that of the September 2015 DSM
for definite comparison of the two DSMs. Before the calculation of the differential rasters, it was necessary to evaluate the systematic errors between pairs of DSMs. The systematic error between each comparison set was determined by selecting more than 10 points at locations that were stable between the two dates and assigning the mean elevation difference at each of

those points to the systematic error (Table 2). The selected comparison points were located on flat areas where little change was observed between the two dates and which were wider than the pixel sizes of the DSMs (Fig. 2).

The comparison results showed that elevations on the January 2007 DSM were higher than those on the September 2015 DSM by 37.9 cm ± 4.5 cm (mean ± standard deviation), and those on the September 2015 DSM were lower than those on the

December 2015 DSM by 20.5 cm ± 2.0 cm. Then, the limit of detection (LoD) of elevation changes for each comparison pair was set to be the twice of the standard deviation of the systematic error between the DSMs, namely, 9.0 cm for the first two periods and 4.0 cm for the last two periods. The topographic changes less than this limit were neglected in the differential rasters and the volume calculation below. The absolute height of the topography was set to the December 2015 DSM values, the GCPs of which had been accurately measured with RTK GNSS. The DSMs were processed with the ArcGIS Raster

Calculator tool to obtain two differential rasters (Figs. 6, 7). The pixel sizes of them were both set to the 1 m for the convenience in volume calculation.

In the inundated areas of the second DSM, apparent elevation changes to the water surface were represented by the first differential raster, rather than the true topographic changes (Fig. 8). The second differential raster showed the water depth there because all inundated areas were no longer existed in December 2015. Thus, the differential rasters were divided into two

figures to separately show the topographic changes (Fig. 6) and elevation changes related to the water surface (Fig. 7). The inundated areas, which looked light brown due to the muddy water, were determined according to the aerial photograph of the second period (Fig. 3b).

## 5 Results and discussion

### 5.1 Comparison of DSMs between January 2007 and September 2015

The levee breach caused changes to the topography up to 300–500 m away from the levee: sand splays were deposited, crevasse channels elongated to the east and then curved southward following the shallow valley but obscured in the downstream were appeared (Figs. 9a, 9b). The crevasse channels formed especially deep, distinguishable ditches in the most upstream parts and structures and other artificial features near the breached levee were damaged or destroyed (Fig. 9c).

Buildings, cars, the prefectural road along the levee, and other artificial features were washed away by the flood and some of

them came to rest within the crevasse channel (Fig. 9d). In the channel behind these obstacles, relatively small, elongated sandy mounds up to 150 m long were deposited. Sand splays were deposited to the north and south of the obstacles, with clear terminations on their downstream sides (Fig. 9b). The southern splays, 150–200 m long and 100 m wide, were larger than the northern ones.

Figure 6a and 7a shows the differential raster between the DSM before (January 2007) and that 3 days after (September 2015)

the flood. Near the levee, the erosional depth exceeded 1 m, decreasing to the east. In the shallow valley formed by the past crevasse channel, however, the topography showed a little aggradation, despite the apparent continuity of the surface seen in

the orthophotos (Fig. 3); the aggradation is probably attributable to the presence of crops before the flood and the submergence of the land surface in September 2015. The thickness of the sand lobes was estimated to be up to 60 cm in the north and 80 cm in the south of the research area, except where small sandy mounds had aggraded behind obstacles on the sand splays. Small sandy mounds behind the obstacles with heights of 20 cm or more are recognizable in the differential raster.

In addition to the obvious topographic changes, some more subtle changes were detected. For example, aggradation of 10 cm or more was widely observed beyond both the north and south sand splays, and in some locations, the deposition exceeded 50 cm. However, because the systematic error between the DSMs was removed by using the elevation of the paved road, the differential raster represents not only changes caused by the 2015 flood but also changes in the status of the agricultural lands, such as the presence of crops and the soil built up over the course of more than eight years. Careful evaluation of such changes

is needed for a more detailed discussion of flood-related changes.

**5.2 Comparison between DSMs of September 2015 and December 2015**

Three months after the flood (December 2015), the crevasse channel was no longer submerged, and the areas of sand deposition appeared lighter in color than other, more muddy areas (Fig. 3c). Other than natural changes, some restoration and repair works by humans were apparent. The breached levee had already been repaired temporarily, the scouring along prefectural road had

been partly filled and leveled, and the road had been reopened. Modification of the immediate post-flood topography was less in distal areas, although some changes are visible in the DSM and orthophoto: for example, crops had grown or been cut in the agricultural lands, and sediments trapped in the canal had been removed along with the obstacles within the crevasse channel. Numerous wheel tracks on the ground indicate that many vehicles had been driven across the study area after the flood, even where marked modification of the topography was not observed. These tracks were obscured in the down-sampled SfM-

derived DSM and the differential raster (Fig. 5).

The differential raster of the DSMs between 3 days after (September 2015) and 3 months after (December 2015) the flood is shown in Fig. 6b and 7b. Significant topographic changes can be observed where restoration works exceeded those completed immediately after the flood. The most remarkable change is the temporary repair of the breached levee, which was raised in height by up to 5.5 m (>1 m in the figure). The area to the east of the repaired prefectural road seemed mostly flat in the

December 2015 DSM and orthophoto, but in fact the amount of fill varied between 10 and 60 cm compared with the original scour topography. In contrast, up to 100 cm of flood deposits were removed to level the ground in the depositional area near the levee. Several buried canals were restored by excavation, and the sand was heaped on both sides of the canals. Although there were some points which experienced local aggradation or degradation due to similar restoration works as such, the crevasse splay was left mostly undisturbed, or had little modification so its features were deformed mainly by natural

phenomena occurring during the 3 months after the flood, except for the reconstruction of the artificial levee and the land filling and leveling along the prefectural road near the breached levee.

The first natural modification was the emergence of the submerged areas of the crevasse channel. The values of the differential raster along the formerly submerged crevasse channel indicate the depth of the pool, because the September 2015 DSM shows mostly the elevation of the water surface in that area. Thus, by using the two differential rasters, the water depth in the crevasse channel 3 days after the flood can be estimated as well as the total amount of erosion of the alluvial ridge. They show that the water depth was 20–60 cm, and erosion of up to 150 cm, with higher values near the levee, was caused by the flood. In the restored areas along the breached levee, however, the true erosional depth of the scouring, certainly the most significant within the study area, could not be determined from the DSMs because the scouring was filled in before the UAV flight was conducted. This is an example of the difficulties faced in the investigation of disasters in developed regions.

In the areas that had been covered with muddy sediments, such as beyond the sand splay in the northern part of the study area, shrubs with heights up to 30 cm grew up especially around the edge of the splay deposits after the flood. In natural floodplains, crevasse splay development promotes the formation of new colonies of vegetation (Florsheim and Mount, 2002; Cahoon et al., 2011). This flood might provide a relationship between the topography of the crevasse splay and the location and the type of such new growth. However, such vegetation was removed during the restoration works

As in previous reports of the topography of crevasse splays (O'Brien and Wells, 1986; Bristow et al., 1999), steep slopes formed at the downstream edges of the crevasse splays in the research area. Three months after their formation, however, the eastern edges of the southern splays had aggraded 10–20 cm, and the western edge of one splay had been lowered by 10 cm, indicating that sand composing the splay had moved eastward. The seasonal wind in winter in Japan, the direction of which coincides with the sand and occasionally exceeds 10 m s$^{-1}$ in speed is possibly responsible for the change combined with the dry wintertime climate of Japan.

## 5.3 Volumetric evaluations of the topographic changes

To get further insights into the 2015 flood event, the volume gains and losses in the research area through the 2015 flood were calculated using the two differential rasters. The range of calculation was limited to the area where the topography was clearly affected by the flood in September 2015 (Fig. 10). Although there were some pixels in which considerable elevation changes were recognized outside the calculation range, the status of crops was mainly responsible for the changes rather than the flood deposits and the erosional scours. The areas used for the buildings were also excluded from the calculation.

Using the first differential raster, the aggradational pixels and the degradational pixels were detected within the calculation range, and the pixels that had smaller values than the LoD of the differential raster were excluded from the calculation (Fig. 10). Then the values of volume gains and losses were separately calculated with the areas and average values of the pixels. Distinguishable sand lobes deposited within the aggradational pixels were a characteristic feature of the crevasse splay, so their volume and area within the entire aggradational pixels were additionally calculated. In some parts of degradational areas, remaining inundated water in the September 2015 DSM had hided the true erosional depths, which could lead to underestimation of the volume loss (Fig. 8). So, the water volume calculated with the differential raster of September 2015

and December 2015 (Fig. 7b), which showed the inundated depths, was added to the volume losses. However, some inundated areas were covered by plants or sand splay deposits and the pixels of the differential raster showed elevation gain, indicating that these areas were not erosive. So, pixels with elevation losses on the first differential raster were only used to calculate the water volume to be added to the volume losses (Fig. 10). Moreover, the inundated depth in the areas which were leveled for the restoration works during the research period could not be estimated with the differential raster. The depth of the scouring was larger near the breached levee, and reached more than 2 m in the maximum (Kanto Regional Development Bureau, Ministry of Land, Infrastructure, Transport and Tourism, 2016). Thus, the water depth of the outer edge of leveled range, which was assumed to be 0.3 m, was substitutionally used to estimate the least water volume there.

Calculation indicated that 78.3% of the entire calculation range marked topographic changes larger than the LoD between the first two periods: 57.4% was aggradational, and 20.9% was degradational (Table 3). The total volume gain was 37 187 m$^3$ and the volume of the distinguishable sand lobes was estimated to be 12 467 m$^3$ of the number, which suggests that deposition of the sand lobes was responsible for 33.5% of the aggradation in volume, while the area they occupied was 24.2% of the all aggradational areas. This gap indicates that the formation of these sand lobes might play a key role in sedimentation during this flood. The volume loss estimated using the first differential raster was 73 948 m$^3$ and the further volume loss of 6 478 m$^3$ represented by the inundating water volume was added to it. Thus, the total erosional volume was estimated to be 80 426 m$^3$, which means that using another DSM in the different period contributed to an improvement by 8.1%.

The distribution of the degraded pixels was almost limited to the proximal of the breached levee while the aggradational pixels were located around the erosional areas and more distant from the breached levee. The aggradational areas were about three times larger than the degraded areas. However, the average value of topographic change in degradational pixels (1.60 m) was quite larger than that of the aggradational pixels (0.27 m) and the total volumetric changes in the calculation range was minus 43 239 m$^3$, suggesting that the 2015 flood event was erosive within the research area.

By December 2015, the post-flood restoration works including reconstruction of the breached levee and leveling of intensively eroded ground near it had proceeded, exterior materials imported to the research area. The volume of those materials was estimated at least 20 636 m$^3$ using the aggradational pixels within the range of these restoration works on the second differential raster (Table 4). Because other restoration activities and natural processes were small in scale of material transportation in the research area, this value represents the topographic changes in the three months after the flood for the most part. Combined with the total volumetric changes by the flood above, it is presumed that the research area experienced volume loss of 22 603 m$^3$.

The volumetric estimation above contain some issues. First, the calculated values might be considerably affected besides topography, such as by vegetation and buildings. The crops on the research area elevated the DSM in September 2015 leading to overestimation of the volume gain and destroyed buildings in the erosive reaches could make the eroded volume unnecessarily larger, if not included. Second, although water depth in inundated areas could be estimated using additional DSM, that of leveled area in December 2015 could not be obtained. However, this drawback may be overcome by more

frequent acquisition of the topographic data, which can be easily done using a UAV and SfM analysis. Third, the simple subtraction procedure to generate the differential rasters assumed non-erosional contacts between the original ground surface before the flood and the flood deposits above. However, it was reported that the boundary had a clear erosional characteristic (Matsumoto et al., 2016), thus the estimated volume gain might be underestimated. Moreover, deposition in the pixels accordingly detected as degradational was neglected, too. Field observations may correct these uncertainties, but it would be difficult to measure thicknesses of eroded beds. Nevertheless, quantitative information derived from a time-series of topographic data fairly represents 3D architecture of a crevasse splay and may enable us, in disastrous situations, to consider the amount of deposits that need to be removed and, conversely, the amount of materials necessary for the construction of embankments or for land filling, In the case of this study, volumetric contribution of sand lobes and land filling to the topography might be suit for these applications.

## 5.4 Recording of landforms with low preservation potential by combined usage of UAV-SfM and LiDAR

In the context of disaster management, high-resolution photos and videos taken by UAVs can help in the development of strategies to take in the event of emergencies (Ezequiel et al., 2014; Erdelj and Natalizio, 2016). Another UAV 'eye', namely, topographic data obtained by SfM photogrammetry, would also be informative, particularly if acquired immediately after the occurrence of a disaster. Of course, pre-flood topographic data are necessary to quantify the degree of erosion and deposition. In Japan, open-access LiDAR data with a resolution of 5 m covering the whole country except for some remote areas is available from the Geospatial Information Authority of Japan (Sato et al., 2010) and can be used for comparison with data obtained after disasters. For example, Saito et al. (2016) utilized a combination of UAV-SfM topographic data obtained after a heavy rainfall in the mountains of southeastern Japan and available pre-hazard LiDAR data to estimate sediment yields of landslides. This study and that of Saito et al. (2016) both took advantage of the availability of UAVs, which because of their low cost and flexible operation, can be effectively used to obtain topographic measurements for use in combination with past LiDAR data to determine topographic differences related to sudden events.

UAVs combined with SfM analysis, can easily be used to obtain additional topographic data on successive dates to produce '4D' topographic models, that is, 3D models showing the evolution of the measurement objectives over time (Gomez and Purdie, 2016). This study achieved this at a very primitive level by using DSMs obtained at three different times to estimate inundation depth, shrub growth, and volumetric estimation of the topographic changes. A similar multitemporal analysis of topography would be possible in other areas in Japan and anywhere else in the world where extensive topographic measurements have been made; thus, this method is expected to become applicable to more and more regions.

Another approach to quantifying morphological changes is archival photogrammetry using SfM methods with past aerial photographs (Bakker and Lane, 2016). Although data sources may be limited in number and resolution, this approach can reconstructing historical events and thus extend the timeline of topography to the period before LiDAR data have been

available. The accumulation of knowledge about the nature of hazardous earth surface processes both currently occurring and in the past is expected to help us to mitigate the damage suffered by people and human society from future disasters.

## 6 Conclusions

In this study, repeated topographic measurement by UAV-SfM photogrammetry and LiDAR, with a resolution on the order of $10^{-2}$ and $10^0$ m, respectively, were successfully used to quantify topographic changes on a floodplain caused by the 2015 flood of the Kinu River, including the formation of a crevasse splay and its post-formation modification by both natural and artificial causes. All three topographic data sets used in this study were surface models, but in most of the research area, they represented the ground elevation because there were few buildings and little vegetation except for agricultural crops. However, these objects might have considerable effects in case of volumetric analysis.

The horizontal accuracy of the DSMs estimated from the positions of building edges showed that the difference in resolution of the DSMs determined the accuracy of the comparisons made by raster calculations. The systematic error between DSMs was assumed to be merely the average of the elevation differences at several selected points. In fact, almost all parts of the road, which were assumed to be stable throughout, were shown to experience small elevation changes (less than 10 cm) in both two comparison pairs, January 2007 versus September 2015 and September 2015 and December 2015. However, very high resolution of the UAV-SfM derived DSM might be inappropriate to compare with the LiDAR derived DSMs with relatively low resolution, due to the trivial relief of the surface model derived from wheel tracks, plants, debris and so on. Down-sampling of the former DSM to 1 m, which was equal to that of LiDAR derived DSM, removed these local fluctuation of the topography, enabling more easy comparison of the surface models acquired by LiDAR. Elevation differences on the order of $10^{-1}$ m were detectable between pairs of DSMs after the removal of the systematic error and the adjusting of the resolution. However, care is needed to account for environmental changes due the difference in the time of acquisition, such as vegetation growth and ground modification by humans.

The topography of the crevasse splay was characterized by the intensive erosion near the breached levee and deposition of the flood deposits, especially lobe-shaped sand mounds occupying 33.5% of the all aggradation, surrounding the erosional areas. Volumetric analysis indicated that the area of the degradation pixels was about three times narrower than the aggradation pixels but the estimated volume loss was more than twice as large as volume gain, resulting in the volume loss of 43 239 $m^3$ in all by the event which was suggested to be quite erosive in the research area. Repeated measurement of the topography allowed the post-formation modifications of the crevasse splay to be quantified. Withsurface models, not only changes to the topography itself but also to vegetation, the inundating water, and artificial modifications for restoration and repair could be quantified. The true erosional depths in the inundated areas could be estimated using the DSM in the third period, resulting in improving by 8.1% of the estimation of the total volume loss. The reconstruction of the breached levee and the leveling of intensively eroded ground near it were major causes of the topographic changes after the occurrence of the flood and the value countervailed 25.6 % of the eroded volume by the flood.

However, the volume estimation in this study have some issues including (i) effects other than the topography such as buildings and agricultural crops, (ii) lack of topographic data showing the entire part of the crevasse splay: partly the erosion depths could not be estimated due to the restoration works, and (iii) ifonly a single process worked at a point, namely, erosion in the aggradational pixels and deposition in the degradational pixels were neglected. Use of topographic models in which buildings and vegetation are removed, more frequent measurement of the topography, which may easily be achieved by the UAV-SfM protocol, and the combination with field observation can improve the estimation.

Low-cost and speedy data acquisition (2 h in the field to cover 0.793 km$^2$ in this study) by UAV can be effectively used, particularly when used in combination with LiDAR data obtained beforehand, to capture topography related to natural disasters with low preservation potential because they are in inhabited regions, where the affected topography is likely to be restored soon after the disaster. Because river floods and the consequent formation of crevasse splays occur in floodplains around the world, the accumulation of similar studies will promote our understanding of the splay formation process and enable damages caused by such flooding to be mitigated. To achieve this goal, however, the preparation and maintenance of an extensive LiDAR database is needed to make possible quantification of topographic changes caused by floods and other disasters by UAV-SfM photogrammetry.

**7 Data availability**

The aerial photos taken from the UAV and SfM-processed DSM are not open to the public for further analysis by the authors. The post-flood LiDAR and orthophotos are provided to the authors with special permission of Aero Asahi Corp. The pre-flood LiDAR data and orthophotos are available with permission from Kanto Regional Development Bureau, Ministry of Land, Infrastructure, Transport and Tourism.

*Author contributions* A. Izumida undertook the processing and the interpretation of the data and the preparation of the manuscript with contributions from all co-authors. S. Uchiyama undertook the data acquisition with the UAV and the SfM processing. T. Sugai designed the research and gave the final approval of the article.

*Competing interests.* The authors declare that they have no conflict of interest.

5    *Acknowledgement.* We gratefully thank Hiroshi Kobayashi of Aero Asahi Corporation for providing the processed DSM derived from the post-flood LiDAR data and the orthophotos acquired in September 2015. This study was financially supported by JSPS KAKENHI Grant Number JP26282078.

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

**Table 1: Error of each GCP and RMSE for the all 15 GCPs in the SfM processing.**

| GCP | X error (cm) | Y error (cm) | Z error (cm) | Total error (cm) |
|---|---|---|---|---|
| G1 | -2.34 | 3.43 | -0.20 | 4.15 |
| G2 | 3.14 | -2.48 | 2.76 | 4.86 |
| G3 | -1.77 | -2.47 | -0.21 | 3.05 |
| G4 | -0.16 | -2.59 | 0.39 | 2.62 |
| G5 | 3.37 | 1.09 | -1.13 | 3.72 |
| G6 | 2.95 | 3.01 | 1.63 | 4.52 |
| G7 | 3.08 | -1.56 | -2.70 | 4.38 |
| G8 | 0.09 | 1.41 | 1.35 | 1.95 |
| G9 | -4.33 | -1.53 | -2.60 | 5.28 |
| G10 | -0.27 | 1.02 | 0.64 | 1.23 |
| G11 | -0.62 | -0.81 | 0.49 | 1.13 |
| G12 | 0.76 | -0.76 | -0.39 | 1.14 |
| G13 | -2.17 | 2.50 | 1.88 | 3.81 |
| G14 | -3.22 | -2.51 | -2.34 | 4.70 |
| G15 | 1.32 | 2.45 | 0.16 | 2.78 |
| RMSE | 2.37 | 2.14 | 1.58 | 3.56 |

**Table 2: Points used to compare each pair of DSMs**

| Comparison pair | Number of points compared | Maximum elevation difference between points (cm) | Minimum elevation difference between points (cm) | Mean elevation difference between points (cm) | Standard deviation of the difference between points (cm) |
|---|---|---|---|---|---|
| Jan 2007–Sep 2015 | 11 | 45.9 | 28.8 | 37.9 | 4.5 |
| Sep 2015–Dec 2015 | 15 | 23.1 | 15.7 | 20.6 | 1.8 |

**Table 3: Results of the volumetric calculations (changes between January 2007 and September 2015).**

| | Aggradation | | Degradation | | | | Total changes in the calculation range between Jan 2007 and Sep 2015 |
|---|---|---|---|---|---|---|---|
| | Sand lobes | Total | Apparent changes between Jan 2007 and Sep 2015 | Inundated area (not leveled in Dec 2015) | Inundated area (leveled in Dec 2015) | Total | |
| Area ($m^2$) | 33334 | 137730 | 50168 | 13105 | 12465 | 50168 | 239747 |
| Area (%) | 13.9 | 57.4 | 20.9 | 5.5 | 5.2 | 20.9 | 100 |
| Volume ($m^3$) | 12467 | 37187 | 73948 | 2739 | 3740 | 80426 | -43239 |
| Average change (m) | 0.37 | 0.27 | 1.47 | 0.21 | (0.3)* | 1.60 | -0.18 |

\* Assumed value for calculation

**Table 4: Results of the volumetric calculations (changes between September 2015 and December 2015).**

|  | Leveled area bewteen Sep and Dec 2015 | Total changes in the calculation range between Sep and Dec 2015 |
|---|---|---|
| Area (m$^2$) | 20331 | 239747 |
| Area (%) | 8.5 | 100 |
| Volume (m$^3$) | 20636 | -22603 |
| Average change (m) | 1.02 | -0.09 |

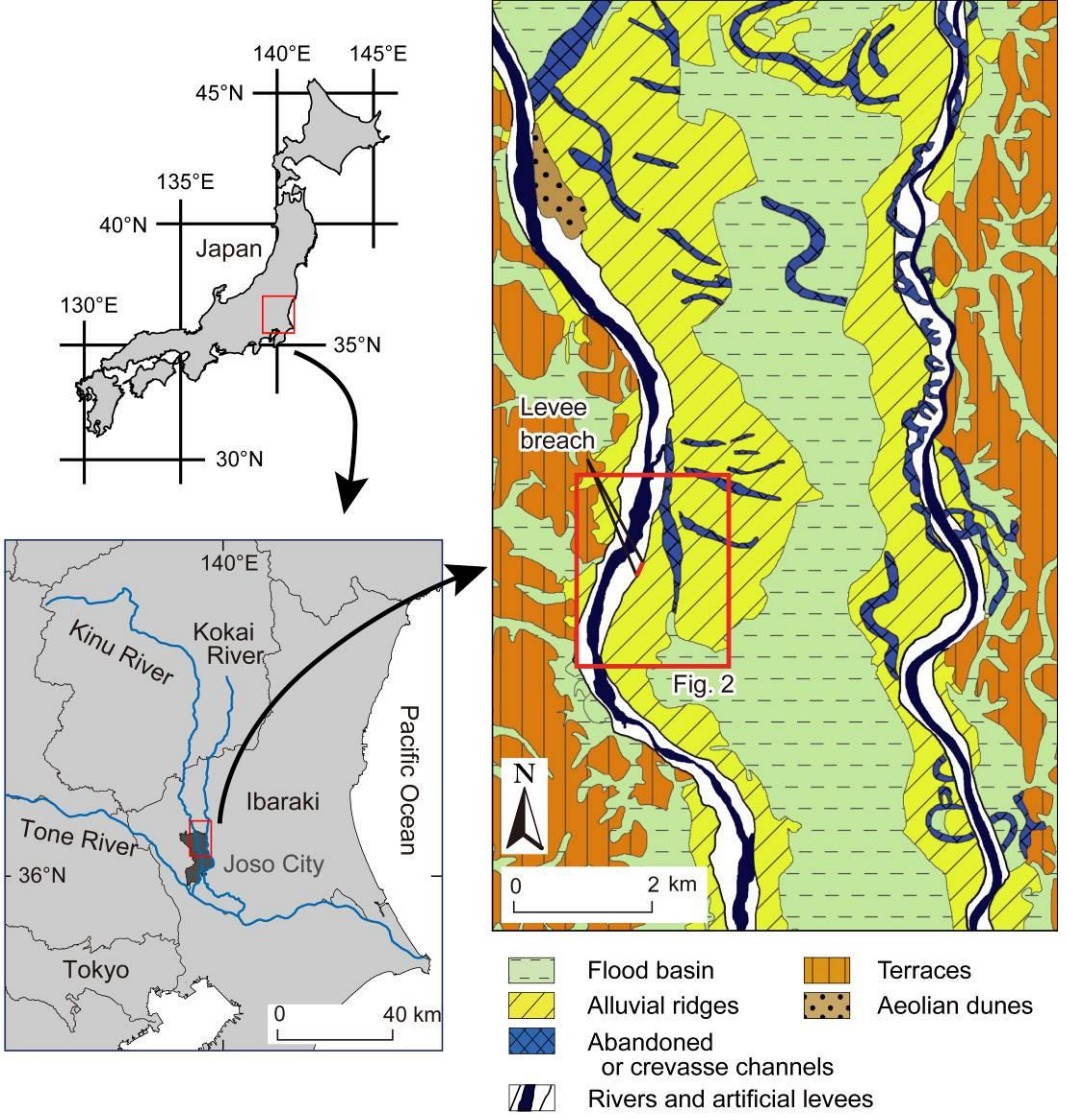

**Figure 1: Location map of Joso City and the Kinu River, and geomorphic map showing the location of the levee breeched during the 2015 flood.**

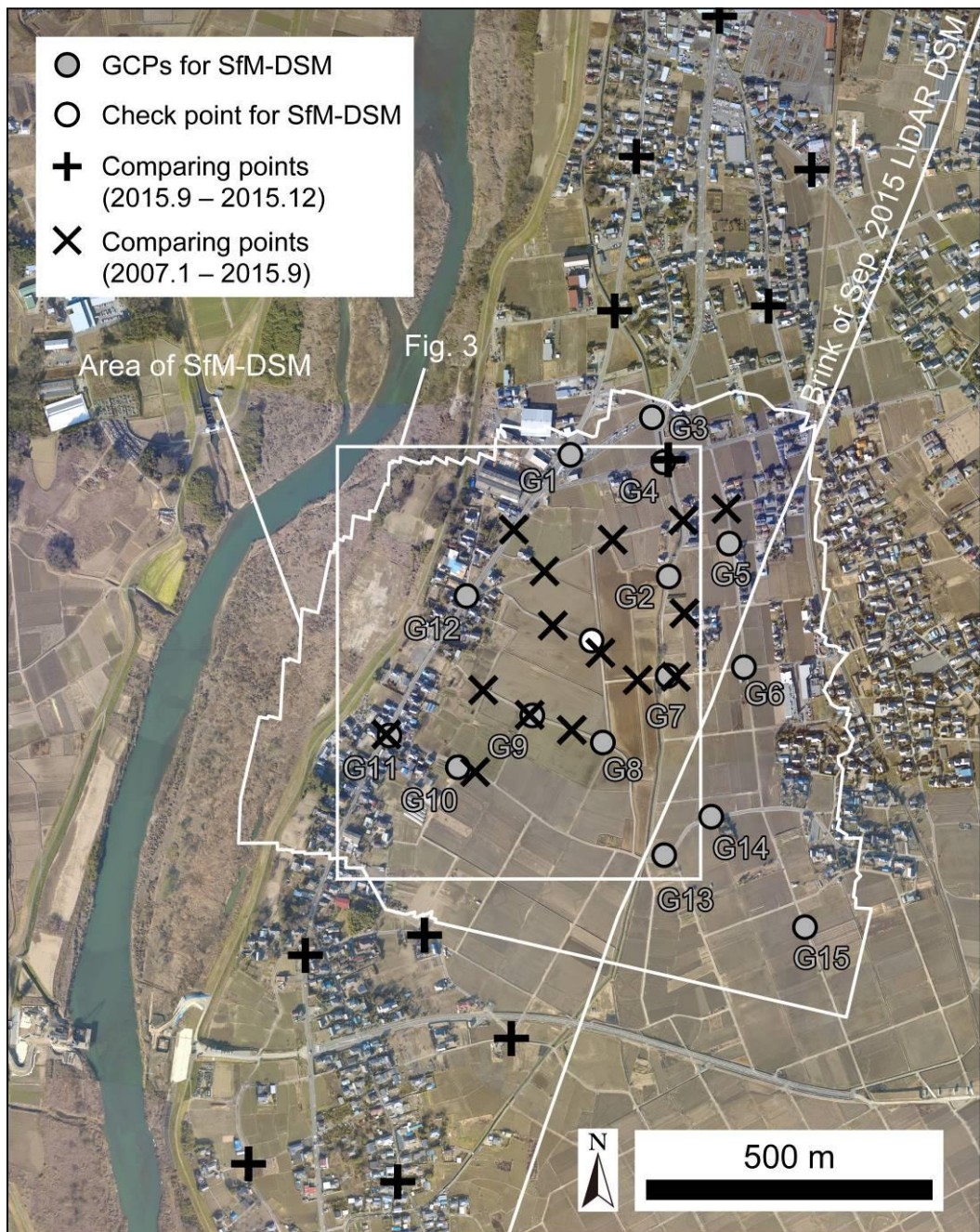

**Figure 2: The distribution of the GCPs and the check point for the SfM-derived DSM and comparing points for each pair of DSM subtraction.**

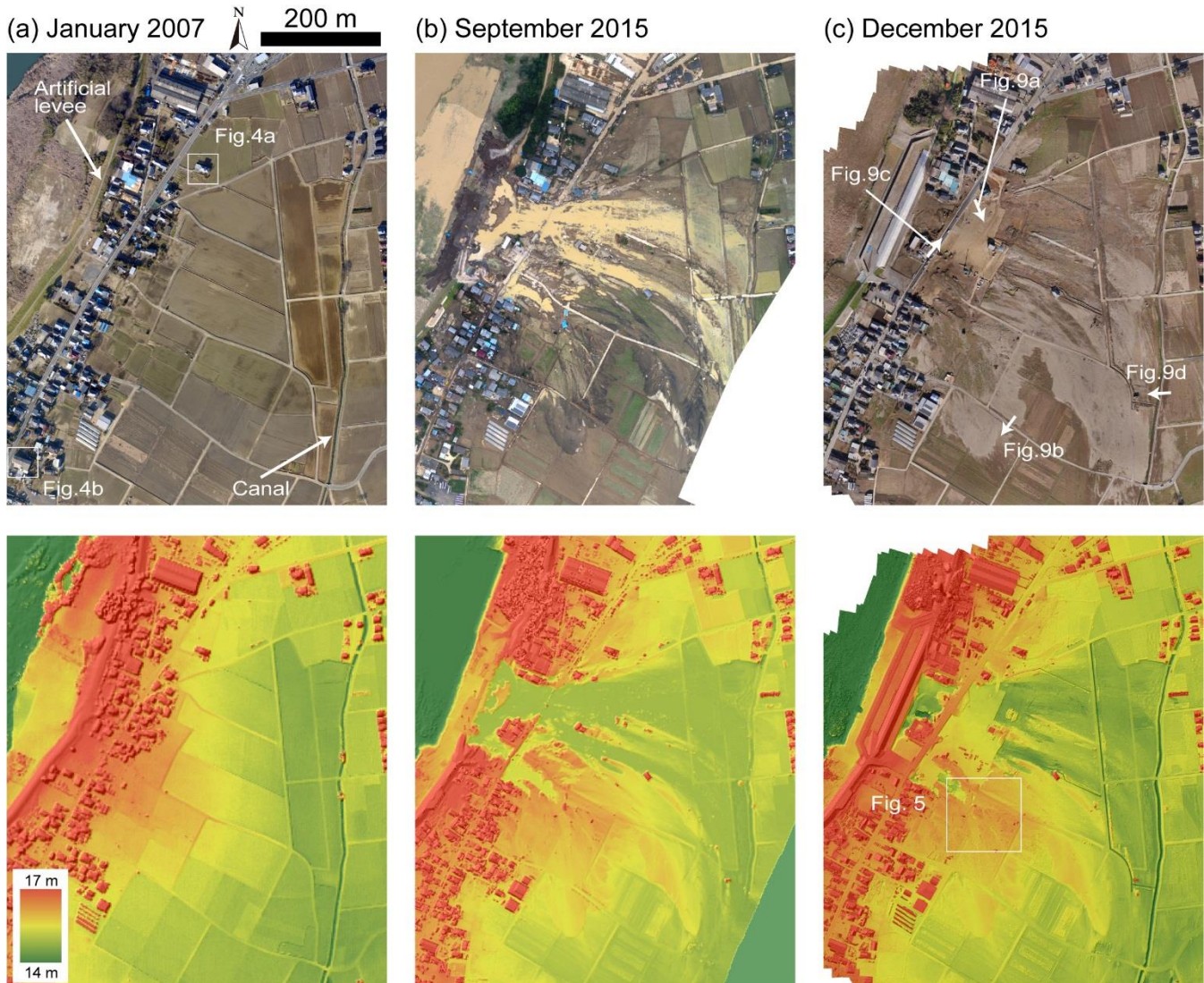

**Figure 3: The three DSMs and orthophotos used in this study. See the text for the information about the acquisition of each data set. Elevations are relative to mean sea level in Tokyo Bay. Note the research area does not correspond to the extent of these DSMs.**

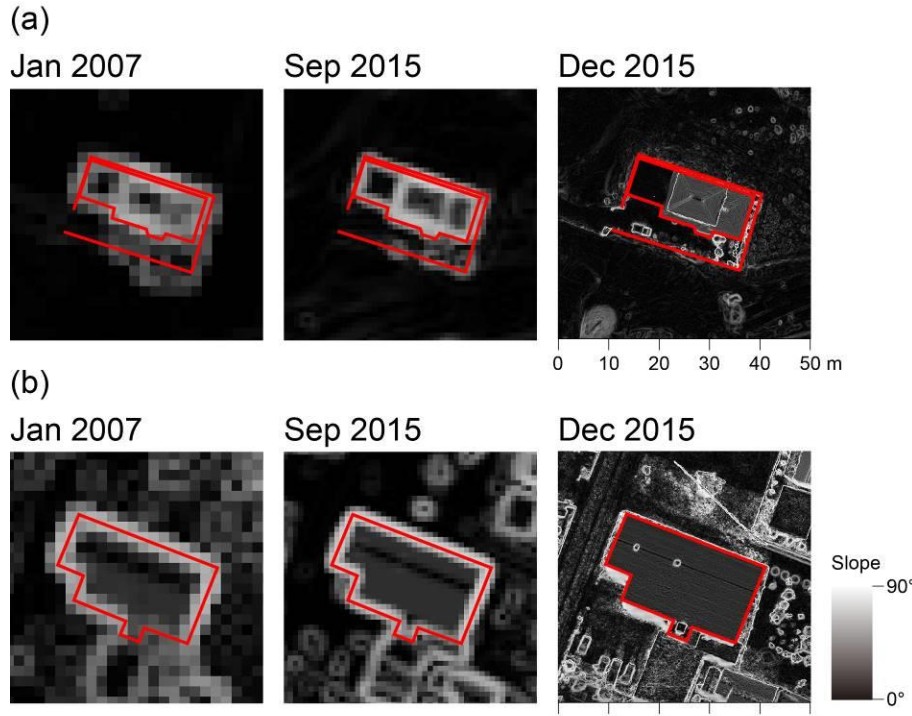

**Figure 4: Gradient maps showing two buildings used to evaluate the horizontal accuracy of the DSMs. The cell sizes are the same as in the originals. Red lines indicate the edges of the buildings or walls on the December 2015 map. Note that these lines are within high slope areas on the maps for the other two dates. See Fig. 3a for the locations.**

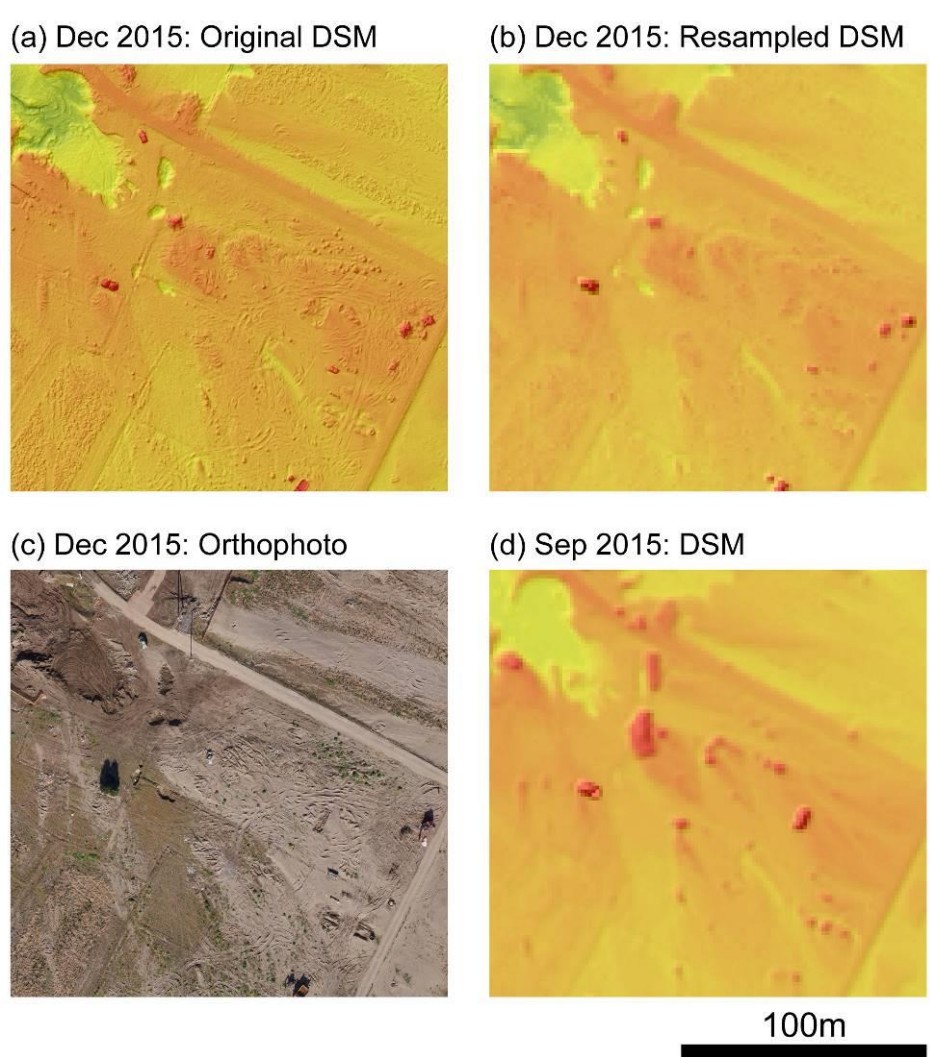

**Figure 5: Enlarged views of Fig. 3 for comparison of the DSMs with different resolutions. (a) Original DSM in December 2015 (resolution: 3.84 cm). (b) Down-sampled DSM in December 2015 (resolution: 1m). (c) Orthophoto in December 2015. (d) DSM in September 2015 (resolution: 1m). Note that the trivial relief in (a) was greatly removed in the down-sampled DSM in (b).**

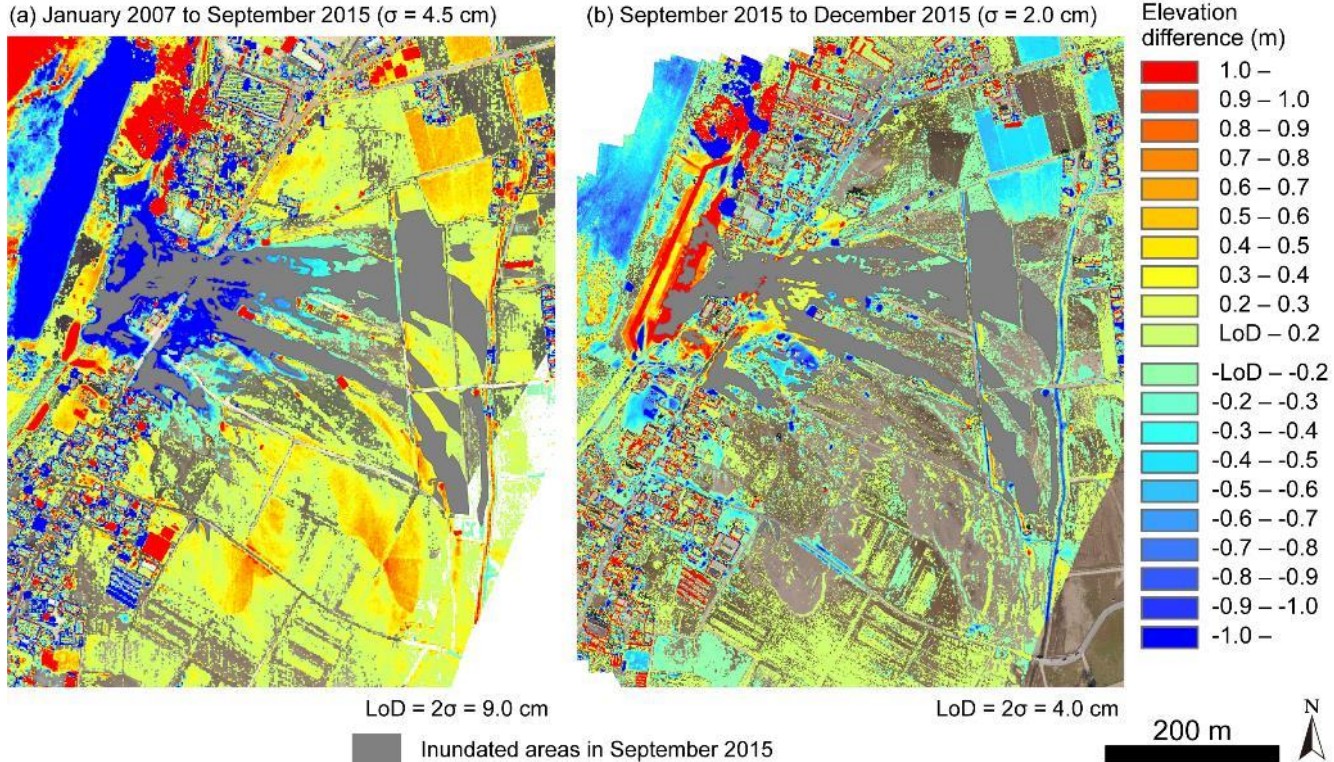

**Figure 6: Differential rasters indicating the elevation differences between two successive snapshots after the removal of the systematic error and the adjustment of resolution. (a) The January 2007 DSM subtracted from the September 2015 DSM. Note that erosion near the breached levee was more than 1 m. (b) The September 2015 DSM subtracted from the December 2015 DSM. Note that inundated areas in period of the second DSM were masked (see Fig. 8 for the elevation changes in these areas).**

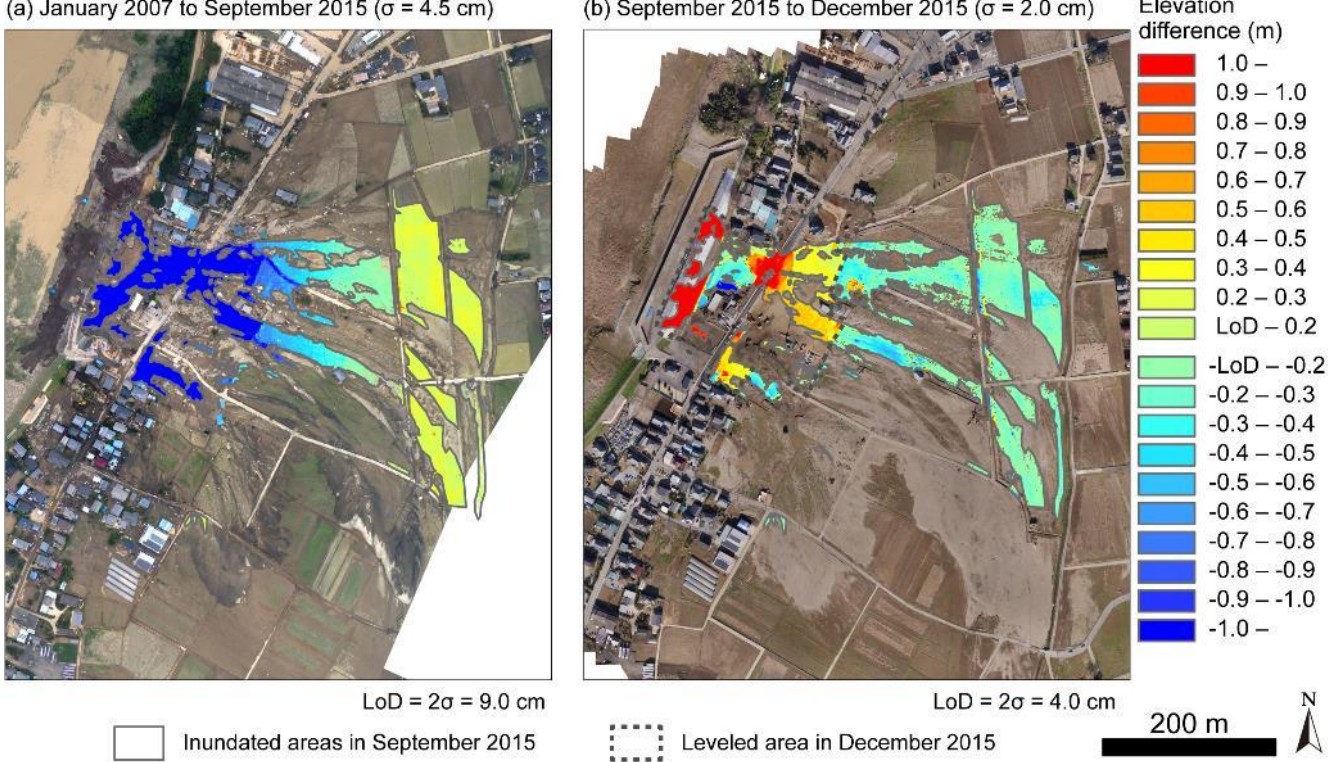

**Figure 7: Differential rasters indicating the elevation differences between two successive snapshots after the removal of the systematic error the adjustment of resolution in the inundated areas in period of the second DSM. (a) The January 2007 DSM subtracted from the September 2015 DSM. Note that erosion near the breached levee was more than 1 m. (b) The September 2015 DSM subtracted from the December 2015 DSM. The elevation changes within the dashed line does not represent the water depths due to the land filling.**

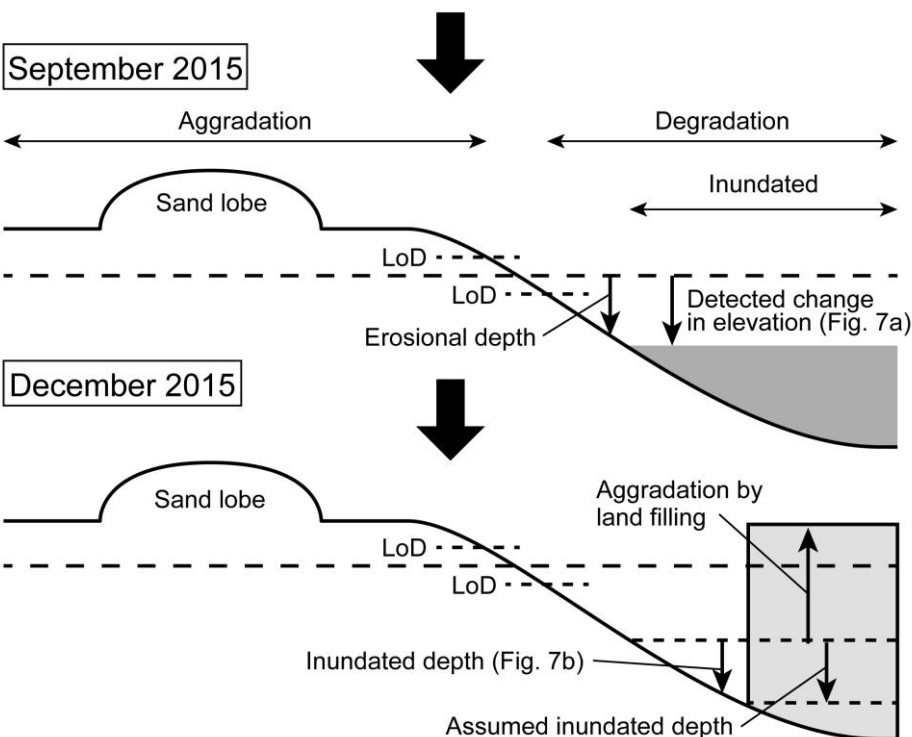

**Figure 8: Schematic description of the topographic changes by the flood and the afterward disappearance of inundated water and landfilling in this study. Note that elevation changes smaller than LoD were neglected in the differential rasters and the volume calculation.**

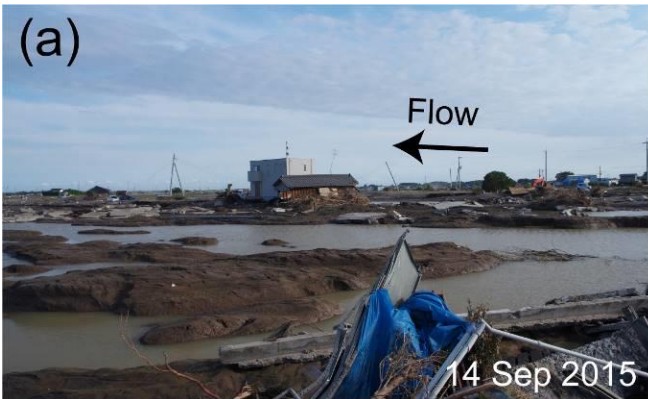
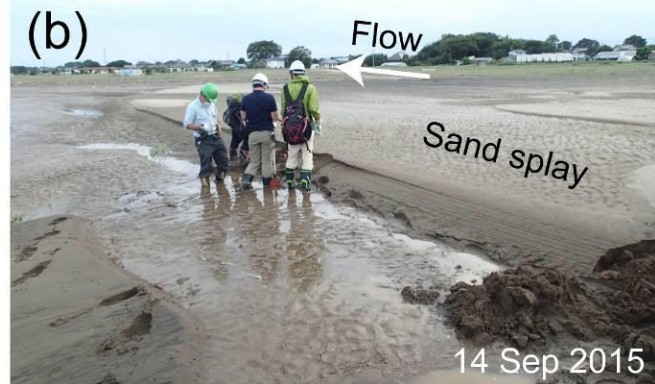
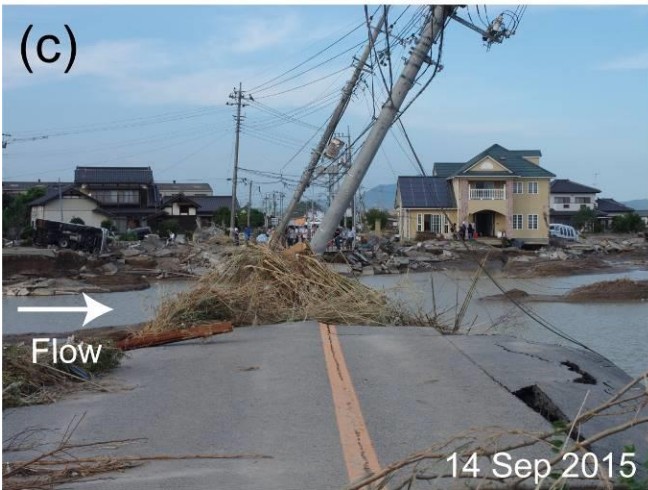
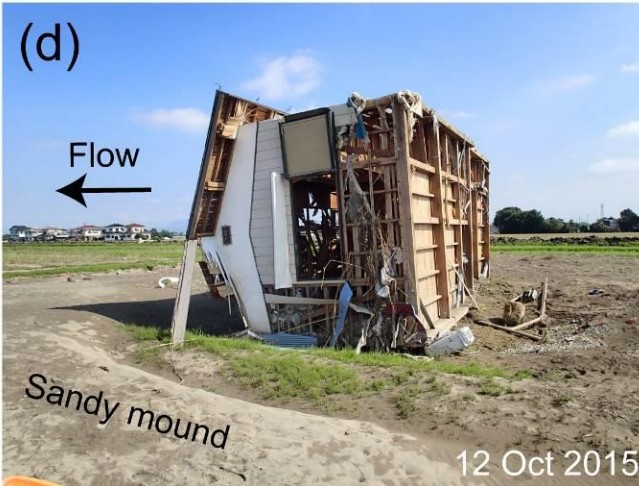

**Figure 9: Photos of the study area. (a) Crevasse channel (breach scouring) near the levee. Note the building still standing after the flood in the center of the photos. (b) Sand splay in the southern part of the study area. (c) Damaged prefectural road beside the breached levee. (d) Overturned and transported house. Note a sandy mound in the downstream direction from the house. See Fig. 3c for the photo locations. Photos were taken by A. Izumida and T. Sugai.**

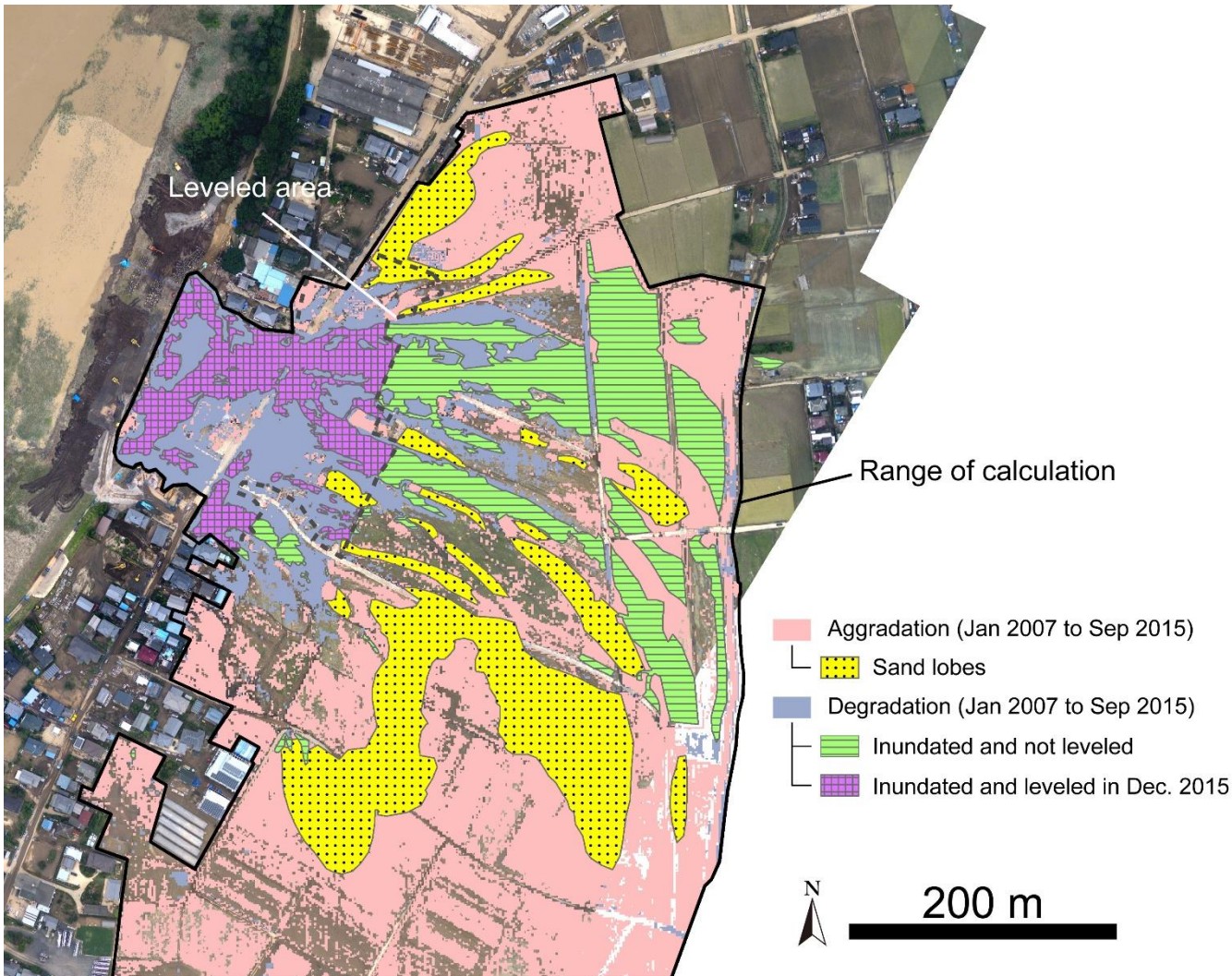

**Figure 10: The calculation range of the volumetric evaluation and topography types used in the calculation. Note that the areas which were inundated but not were leveled in December 2015 (green) contain aggradational pixels, which were not used for the estimation of the water volume (but included in the volume gain).**

