# Peer review of "Application of UAV-SfM photogrammetry and aerial LiDAR to a disastrous flood: repeated topographic measurement of a newly formed crevasse splay of the Kinu River, central Japan"

_Natural Hazards and Earth System Sciences, 2017_

## Referee Comment (RC1) · Anonymous Referee #1 · 27 Feb 2017

GENERAL COMMENTS

This study gives a nice insight in the topographic changes that occurred during a levee breach / crevasse splay and its aftermath. Taking the analysis of this data set slightly further, notably through determining volume changes, would improve the paper significantly and give more insight in the events.

SPECIFIC COMMENTS

Title - You mention 'disastrous flood' and 'disaster' in the paper. It would be good to give some indication of the magnitude of this event in terms of return time, wounded /

casualties and costs.

2-2/3 Unclear what you mean here: 'land use' and 'human-built structures'. Do you want to indicate that breaches differ when there is agriculture/roads as opposed to natural floodplain vegetation? Or do you want to indicate that lobes of sediment accumulate behind human objects such as buildings? Or otherwise?

2-23 I think you should make the nuance here that the topography before a flood may be available but not of sufficient detail to investigate the topographic change.

3-27/28 Can you also indicate peak (and average) discharge and flood return time?

In general the text concerning the STUDY AREA and THE 2015 FLOOD OF THE KINU RIVER can be shortened and more to the point. On the other hand, relevant information on settlements, land-use and the flood impact (wounded/casualties and costs) should be mentioned here.

5-13 Indicate the locations / distribution of GCPs in a figure.

5-12 What were the results for the automatic camera calibration, did it return the known values?

5-24 If you mention this, give an indication for how many locations and how they are distributed. Did you also include houses on the top-right (figures 2a-c)?

6-2/4 How are these points distributed, with respect to each other and the GCPs. Include these in a figure.

6-9/19 You can calculate a limit of detection using these numbers and apply this in the figures.

6-13/14 Why didn't you use the lower of the two resolutions? Using the higher resolution can lead to local scale effects.

For the applied systematic error correction you assume that the error is both linear and

in the Z direction. Therefore it is important to: 1) show the distribution of the GCPs and control points (see earlier comments) and 2) show that there is no doming in the SfM DEM - this is a known problem see e.g. James and Robson (2014). James, M. R. and Robson, S. (2014), Mitigating systematic error in topographic models derived from UAV and ground-based image networks. Earth Surf. Process. Landforms, 39: 1413–1420. doi:10.1002/esp.3609

7-1 I think you should exclude inundated areas in the DEM of difference (either filter these areas out indicate them using a different color) - you could include them in a separate figure to indicate the water depth.

7-18 See earlier remark on limit of detection.

8-8/9 Please support these comments with volume calculations.

8-21/23 Unclear what you mean with 'important environmental component of the flood-plain' and the relation with human changes. Do you mean that without human impact there would have been more vegetation in the floodplain and the crevasse splay topography might have been different (but how)?

8-31 Is the wind velocity sufficient to pick up the (fine) sediment?

9-7/9 What do you exactly mean with 'simpler topography'? Would it have been possible if the topography was more 'complex' at the later time?

9-21/22 Note that it may also be valuable to research historical events using archival photogrammetry, e.g. Bakker and Lane (2016). Bakker, M., and Lane, S. N. (2016), Archival photogrammetric analysis of river–floodplain systems using Structure from Motion (SfM) methods. Earth Surf. Process. Landforms, doi: 10.1002/esp.4085. 9-31/34 This is a very important point and I think you have a good data set to include these calculations!

Most important remark concerning the RESULTS AND DISCUSSION is that a volumetric analysis would be very valuable (for the area as a whole and / or for certain

regions), indicating the net amount of sediment that came from the river channel, the amount of erosion / sedimentation on the levee/floodplain, the amount of sediment that was redistributed / imported during post-flood works.

TECHNICAL CORRECTIONS

Title - You mention 'multitemporal', but in this case (3 measurements) 'repeated' is perhaps more appropriate.

1-14 'by subtraction' can be removed.

1-16 'carried out by people' can be removed.

1-17 'with different resolutions and acquisition periods': I would say different spatial and temporal resolutions. (Acquisition period can be interpreted as the duration of acquisition, e.g. the flight time of the UAV).

1-18 'sudden' can be removed.

2-4 It is questionable if you should include a reference to this unpublished paper. Both here and further in the manuscript this reference is not required / of added value. I would advise not to include it.

2-4 'Thus' can be removed (there is no direct link with the previous sentence).

3-3 Refer to the figure here.

4-22 Brackets can be removed here and later on when mentioning points/m2.

4-28 Include that this was converted to raster (similar to the pre-flood lidar data).

5-8/10 Only mention the usable photos.

6-17/27 This is a (specific) description of the study area - not results.

7-1 This part of the method.

Table 1: This table can be removed (little information and no additional information than

in the text).

Table 3: What is the (estimated) velocity required to transport sediment? Mean velocity is probably not a suitable measure, perhaps the 90th percentile(?).

Figure 1: Include 'Japan' in the left top figure.

---

## Author Comment (AC1) · 23 Mar 2017

Authors: Thank you for your interests about our paper and valuable comments to improve it. We would like to respond your comments in point-per-point manner. We are afraid that the page/line numbers the referee indicated are different from those of our discussion paper. We have inferred the corresponding texts from the referee's comments, but we're sorry if there were any mismatch.

GENERAL COMMENTS This study gives a nice insight in the topographic changes that occurred during a levee breach / crevasse splay and its aftermath. Taking the

analysis of this data set slightly further, notably through determining volume changes, would improve the paper significantly and give more insight in the events.

Authors: Thank you for commenting and suggesting for volume calculation. We are also sure that volume calculation will effectively describe this event and would like to do that. Volume calculations are intended to represent the volume of sediments that were transported from the river and that of materials that were eroded and removed from the research area. and will be based on the difference between the first two DSMs (Jan 2007 and Sep 2015). However, September 2015 DSM has inundated areas, which can lead to underestimate of the eroded volume, so the December 2015 DSM will be used to estimate the correct erosion depths.

SPECIFIC COMMENTS Title - You mention 'disastrous flood' and 'disaster' in the paper. It would be good to give some indication of the magnitude of this event in terms of return time, wounded / casualties and costs.

Authors: Thank you for suggesting additional information that is needed for our paper. The "disastrous" aspect of the flood and rainfall will be mentioned in the revised form (see our reply below).

2-2/3 Unclear what you mean here: 'land use' and 'human-built structures'. Do you want to indicate that breaches differ when there is agriculture/roads as opposed to natural floodplain vegetation? Or do you want to indicate that lobes of sediment accumulate behind human objects such as buildings? Or otherwise?

Authors: Thank you for commenting. We believe that the artificial modification of the floodplain causes both of what the referee suggested. Both are considered to be complexities added to crevasse splays by human works. In the revised manuscript, more explanation will be added.

2-23 I think you should make the nuance here that the topography before a flood may be available but not of sufficient detail to investigate the topographic change.

Authors: Thank you for suggestion. Actually, in many cases it is difficult to quantify the topographic changes in much detain even if the data before a event exists because of its low resolution. However, high-resolution LiDAR data that covers almost the nation is available in Japan—which makes researches like ours possible. The rarity of using a rich data will be mentioned in the revised manuscript.

3-27/28 Can you also indicate peak (and average) discharge and flood return time?

Authors: Kinugawa-Mitsukaido gauge station, which is at 10 km downstream of the breached point of the Kinu River marked a peak discharge of c.a. 4000 m3s-1, which is the maximum observed in history of 90 years, and a similar discharge (c.a. 3900 m3s-1) occurred only once at 1949 there (KRDB, 2015). Unfortunately, average discharge during the flood was unclear for the authors. Statistical analysis by Yoshimura et al. (2016) suggests that the return time of the cumulative rainfall for a single day, two days and three days over the drainage area of the Kinu River during the 2015 flood is 95, 138 and 237 years, respectively. These information will be added in the revised paper.

(referee) In general the text concerning the STUDY AREA and THE 2015 FLOOD OF THE KINU RIVER can be shortened and more to the point. On the other hand, relevant information on settlements, land-use and the flood impact (wounded/casualties and costs) should be mentioned here.

Authors: Thank you for suggestion. Actually, some information about the river and region is not directly related to the scope of the manuscript. It will be omitted in the revised version. Human suffering in Joso City, where most of damages by the levee breach of the Kinu River occurred, was 2 deaths and 44 injured (Joso City, 2016). The economic loss by the flood and rainfall have been provisionally estimated to be 155.5 billion yens in Ibaraki Prefecture and 289.6 billion yens for the entire disaster of September 2015 (MLIT, 2016). These information will be added to the revised manuscript. See the report of Nagumo et al. (2016) for more detailed damages by the 2015 flood of the Kinu River.

5-13 Indicate the locations / distribution of GCPs in a figure.

Authors: A new figure that shows locations of GCPs and comparing points for DSM subtraction will be added to the revised manuscript as they are distributed inside and outside of the Fig. 2.

5-12 What were the results for the automatic camera calibration, did it return the known values?

Authors: Thank you for commenting. We have not conducted a camera calibration manually for the GR used in this study before, so it might be nonsense to show the results of automatic camera calibration. However, we believe that the obtained DSM successfully show the trend of topographic changes at least in 10-1 order, as explained in the discussion paper.

5-24 If you mention this, give an indication for how many locations and how they are distributed. Did you also include houses on the top-right (figures 2a-c)?

Authors: All the three DSMs used in this study were georeferenced with accurate GNSS systems, so we believe that the two locations shown in Fig. 4 are sufficient to check the horizontal accuracy. The sentence "The same $\sim$ (5-14)" will be omitted in the revised version.

6-2/4 How are these points distributed, with respect to each other and the GCPs. Include these in a figure.

Authors: See the reply above.

6-9/19 You can calculate a limit of detection using these numbers and apply this in the figures.

Authors: Thank you for suggestion. We have set a limit of detection of elevation changes for each comparison pair as the twice of the standard deviation of the systematic error between the DSMs, namely, 9.0 cm for the first two periods and 4.0 cm for

the last two periods (newly calculated value; see the reply below). These values will be applied to the Fig. 5 to indicate the location where significant topographic changes occurred. Volumetric calculations will be conducted only where the topographic changes were larger than this limits.

6-13/14 Why didn't you use the lower of the two resolutions? Using the higher resolution can lead to local scale effects.

Authors: Thank you for comment. We tried to downsample the SfM-DSM to 1 m resolution which is equal to that of LiDAR data in September 2015, and as a result, we have decided to use the lower resolution for raster calculation in our revised manuscript. Although the figures look more elaborated with the original resolution, 1 m resolution is enough to represent the artificial and natural changes of crevasse splay to some extent, and trivial relief of the surface model derived from wheel tracks, plants, etc. have been significantly removed with that resolution. However, the higher resolution (1 m) have been used in the DoD for the first two periods (January 2007 and September 2015) because a significant effect has not been obtained with lower resolution. In the revised manuscript, relevant texts and figures will be modified.

(referee) For the applied systematic error correction you assume that the error is both linear and in the Z direction. Therefore it is important to: 1) show the distribution of the GCPs and control points (see earlier comments) and 2) show that there is no doming in the SfM DEM - this is a known problem see e.g. James and Robson (2014). James, M. R. and Robson, S. (2014), Mitigating systematic error in topographic models derived from UAV and ground-based image networks. Earth Surf. Process. Landforms, 39: 1413–1420. doi:10.1002/esp.3609

Authors: Thank you for comment. 1) See the reply above. 2) In this study, we have used the following technics to prevent the dooming effect.  Placed sufficient number of GCPs.  Not used an automated camera gimbal, thus controlled vertical photos were not taken.  The camera was flexibly held by a picavet (Inoue et al., 2014, Fig.

11), so off-nadir angles a little varied for each photo.  Off-nadir angle was between 10–15 degrees backward. These explanation will be added to the revised manuscript.

In addition, a GNSS survey point which was not used as a GCP for SfM processing confirmed there is no dooming in the DSM. The suggested reference will be cited in the revised version of the manuscript for short explanation of the dooming effect.

7-1 I think you should exclude inundated areas in the DEM of difference (either filter these areas out indicate them using a different color) - you could include them in a separate figure to indicate the water depth.

Authors: Thank you for suggestion and we agree with the referee. To indicate the topographic changes and inundation depths separately, a new figure will be added and inundated area (on 13 September 2015) in Fig. 5 will be masked by a different color. The inundated area have been determined mainly by interpretation of the aerial photographs used in this study (inundated water was muddy in the photograph in September 2015, resolution: 0.25 m). The orthophoto obtained by the UAV have been secondary used.

7-18 See earlier remark on limit of detection.

Authors: The limit was set to the twice of the standard deviation of the systematic error. The non-transparent cells in the Fig. 5 will be subjected to the description of the topographic changes and the volume calculation.

8-8/9 Please support these comments with volume calculations.

Authors: Thank you for commenting. When the discussion paper was written, we had considered the flood sediments in the canal to be non-negligible compared to the entire volume of sediments. However, the volume calculation have revealed that is not the case. In the revised manuscript, that text will be omitted, and instead, the entire mass transportation will be discussed.

8-21/23 Unclear what you mean with 'important environmental component of the floodplain' and the relation with human changes. Do you mean that without human impact there would have been more vegetation in the floodplain and the crevasse splay topography might have been different (but how)?

Authors: In floodplains of natural rivers, the type and distribution of vegetation can be affected by emergence of crevasse splays by providing new deposits and creating new topography. In the 2015 flood, however, new vegetation was lost through post-disaster restoration works, although that vegetation did exist around the sandy mound of crevasse splays in December 2015. They could potentially respond and record the nature of a crevasse splay and the adjacent floodplain if they were not removed by human. In the revised manuscript, relevant texts will be rewritten.

8-31 Is the wind velocity sufficient to pick up the (fine) sediment?

Authors: In Joso City, the maximum wind velocity in a day sometimes exceeds 10 m s-1 in winter, so it is possible to think the wind is responsible for the post-deposition deformation of the crevasse splay. However, the analysis of the wind process is not the scope of this study, and we believe that this topic should not be examined in detail. Thus, Table 3 and Figs. 6c and 6d will be omitted in the revised manuscript, and the possibility of the wind process will be shortly mentioned in the revised text.

9-7/9 What do you exactly mean with 'simpler topography'? Would it have been possible if the topography was more 'complex' at the later time?

Authors: Thank you for commenting. Within the research area, the land use was almost cultivated land before the flood, so the relief was so small that a resolution of 1 m was enough to represent the ground surface. In contrast, the topography had much relief due to local disturbance of the flood flow and transported deposits. However, we have decided to use a common resolution of 1 m for DSM calculations in the revised manuscript, so these mentions will make little sense. The entire paragraph will be omitted or rewritten.

9-21/22 Note that it may also be valuable to research historical events using archival photogrammetry, e.g. Bakker and Lane (2016). Bakker, M., and Lane, S. N. (2016), Archival photogrammetric analysis of river–floodplain systems using Structure from Motion (SfM) methods. Earth Surf. Process. Landforms, doi: 10.1002/esp.4085.

Authors: thank you for suggesting a valuable paper which have discussed the accuracy of SfM photogrammetry using historical aerial photographs with limited number and resolution. We would like to include it in our reference list as an example of recent studies about reconstructing floodplain topography with SfM photogrammetry. However, our dataset has much more number of photographs for SfM photogrammetry and accurate GCPs, and georeferenced LiDAR DSMs, so it might be difficult to directly compare the 2015 flood with historical events with archival photogrammetry. In the revised version, the possibility of reconstructing the historical events will be mentioned.

9-31/34 This is a very important point and I think you have a good data set to include these calculations!

Authors: Thank you for commenting. We think the sandy lobes deposited in the research area can be a good example because they are easy to detect and represent considerable portion of sediments brought to the research area by the flood. This calculation will be additionally conducted to make more specific insight into this event.

(referee) Most important remark concerning the RESULTS AND DISCUSSION is that a volumetric analysis would be very valuable (for the area as a whole and / or for certain regions), indicating the net amount of sediment that came from the river channel, the amount of erosion / sedimentation on the levee/floodplain, the amount of sediment that was redistributed / imported during post-flood works.

TECHNICAL CORRECTIONS

Authors: Thank you for the corrections. These comments will be reflected in the revised manuscript.

Title - You mention 'multitemporal', but in this case (3 measurements) 'repeated' is perhaps more appropriate.

Authors: 'multitemporal' have been replaced by 'repeated' in the title.

1-14 'by subtraction' can be removed.

Authors: 'by subtraction' have been removed.

1-16 'carried out by people' can be removed.

Authors: 'carried out by people' have been removed.

1-17 'with different resolutions and acquisition periods': I would say different spatial and temporal resolutions. (Acquisition period can be interpreted as the duration of acquisition, e.g. the flight time of the UAV).

Authors: The text have been rephrased: 'with different spatial and temporal resolutions'

1-18 'sudden' can be removed.

Authors: 'sudden' have been removed.

2-4 It is questionable if you should include a reference to this unpublished paper. Both here and further in the manuscript this reference is not required / of added value. I would advise not to include it.

Authors: That unpublished paper have been removed from the reference list.

2-4 'Thus' can be removed (there is no direct link with the previous sentence).

Authors: 'Thus' have been removed.

3-3 Refer to the figure here.

Authors: Figure 1 have been referred.

4-22 Brackets can be removed here and later on when mentioning points/m2.

Authors: They have been removed.

4-28 Include that this was converted to raster (similar to the pre-flood lidar data).

Authors: That have been included.

5-8/10 Only mention the usable photos.

Authors: The word '1433 photos' have been removed.

6-17/27 This is a (specific) description of the study area - not results.

Authors: This part have been moved to the study area section.

7-1 This part of the method.

Authors: It have been moved to the method section.

Table 1: This table can be removed (little information and no additional information than in the text).

Authors: It have been removed.

Table 3: What is the (estimated) velocity required to transport sediment? Mean velocity is probably not a suitable measure, perhaps the 90th percentile(?).

Authors: This table have been removed because the wind process is not the point of our study (see the reply above).

Figure 1: Include 'Japan' in the left top figure.

Authors: 'Japan' have been included.

---

## Referee Comment (RC2) · Y. S. Hayakawa (Referee) · 3 May 2017

**[ I post this comment on behalf of the reviewer ]**

GENERAL COMMENTS

This study is very interesting, because it shows the detail of topographic changes that occurred during a levee breach / crevasse splay. And it shows the usefulness of UAV-SfM method. I judge that this study is worth for publication in this journal.

TECHNICAL CORRECTIONS

P4 20-21 Kanto Regional Development Bureau, Land, Infrastructure and Transportation Ministry of Japan
P10 29-30 Kanto Regional Development Bureau, Ministry of Land, Infrastructure, Transport and Tourism
Right indication Kanto Regional Development Bureau, Ministry of Land, Infrastructure, Transport and Tourism

---

## Author Response (AR1)

**Final response to the Referees' comments**

Authors: Thank you for your interests about our paper and valuable comments to improve it. We would like to respond to the comments in point-per-point manner. According to the comments, we have made some modifications to the manuscript and added new references. Please check our response to the comments and marked-up manuscript below.

**Editor's comment**

Comment 1: One minor comment from the editor: concerning the comment 5-12, if not all the resultant parameters of automatic camera calibration, please provide the reprojection error values at least.

Authors: page 6, lines 20–21 The texts below have been added.
*"The reprojection error was 0.455 pixel, which was no larger than that of similar SfM analysis with images taken by UAV (Pineux et al., 2016)."*

**Referee 1**

**GENERAL COMMENTS**

This study gives a nice insight in the topographic changes that occurred during a levee breach / crevasse splay and its aftermath. Taking the analysis of this data set slightly further, notably through determining volume changes, would improve the paper significantly and give more insight in the events.

Authors: Thank you for commenting and suggesting for volume calculation. Section *5.3 Volumetric evaluations of the topographic changes* have been added in the revised manuscript to obtain further insights into the 2015 flood event of the Kinu River. Other than Section 5.3, abstract and Conclusion sections of the revised manuscript have had significant modification including revisions suggested in other comments. See and find each modification in our responses to the Referee Comments below.

**SPECIFIC COMMENTS**

Comment 1 (Title) - You mention 'disastrous flood' and 'disaster' in the paper. It would be good to give some indication of the magnitude of this event in terms of return time, wounded / casualties and costs.

Authors: The texts regarding to the magnitude of the flood have been added modified in the "**3 The 2015 flood of the Kinu River**" section as below.

Page 4, lines 11–13: The text below has been added.
*"Statistical analysis by Yoshimura et al. (2016) suggests that the return time of the cumulative rainfall for a single day, two days and three days over the drainage area of the Kinu River during the 2015 flood is 95, 138 and 237 years,*

*respectively.*

Page 4, lines 15–18: The text below has been added.

*"Kinugawa-Mitsukaido gauge station, which is at 10 km downstream of the breached point of the Kinu River marked a peak discharge of c.a. 4000 m3 s-1, which is the maximum observed in history of 90 years, and a similar discharge (c.a. 3900 m3 s-1) occurred only once at 1949 there (Kanto Regional Development Bureau, Land, Infrastructure and Transportation Ministry of Japan, 2015)."*

Page 4, lines 23–25: The text below,

*"there were 6000 of the 65,000 inhabitants of Joso City had to be evacuated"*

has been modified to,

*"there were two deaths, 44 injured, and 6000 evacuees of the 65,000 inhabitants of Joso City"*

Page 4, line 30–page 5, line 2: The text below has been added.

*"The economic loss by the flood and rainfall have been estimated to be 159.2 billion yens in Ibaraki Prefecture and 294.1 billion yens for the entire disaster by the heavy rain in September 2015 (Land, Infrastructure and Transportation Ministry of Japan, 2017). For further information of 2015 flood of the Kinu River, see Nagumo et al. (2016) which have provided a detailed report of damages and social effects in the flooded areas."*
* * *
Comment 2 (2-2/3): Unclear what you mean here: 'land use' and 'human-built structures'. Do you want to indicate that breaches differ when there is agriculture/roads as opposed to natural floodplain vegetation? Or do you want to indicate that lobes of sediment accumulate behind human objects such as buildings? Or otherwise?

Authors: Page 2, lines 4–7 The text,

*"Present-day levee breaches along rivers under human control create landforms quite different from and, in some cases, much larger than those created by levee breaches along natural rivers, because of artificial land uses (e.g., human-built structures) along rivers (Nelson and Leclair, 2006)"*

has been modified as below.

*"Present-day levee breaches along rivers under human control create landforms quite different from and, in some cases, much larger than those created by levee breaches along natural rivers, because of houses and paved streets which control the distribution of sand lobes by acting as channels of flood flow (Nelson and Leclair, 2006)"*
* * *
Comment 3 (2-23): I think you should make the nuance here that the topography before a flood may be available but not of sufficient detail to investigate the topographic change.

Authors: Page 2, line 27–28

*"documentation of the topography before a flood event is not available"*

has been modified as below

*"reliable topographic data before a flood event which enables the investigation of topographic changes is not available"*
* * *
Comment 4 (3-27/28): Can you also indicate peak (and average) discharge and flood return time?

Authors: See our response to Comment 1.
* * *
Comment 5: In general the text concerning the STUDY AREA and THE 2015 FLOOD OF THE KINU RIVER can be shortened and more to the point. On the other hand, relevant information on settlements, land-use and the flood impact (wounded/casualties and costs) should be mentioned here.

Authors: The texts have been revised as below. Also see our response to Comment 1.

Page 3, line 16: The text below has been added.
*"and their abandoned channels are"*

Page 3, lines 19–20: The text below has been removed.
*"Older crevasse channels dissect the wide alluvial ridges by 1 m deep, suggesting that the Kinu River has flooded repeatedly in this region (Sadakata, 1971)"*

Page 3, lines 20–23: The text below has been added.
*"The alluvial ridges in this floodplain are densely inhabited and most villages are located on them. Other part of the alluvial ridges are used as agricultural lands, predominantly paddy fields. On the other hand, irrigation canals are entirely equipped in the flood basin and abandoned channels, and almost all area are used as paddy fields."*
* * *
Comments 6 (5-13): Indicate the locations / distribution of GCPs in a figure.

Authors: New figure (Fig. 2) and table (Table 1) have been added in page 21 and 27.
* * *
Comments 7 (5-12) What were the results for the automatic camera calibration, did it return the known values?

Authors: Thank you for commenting. We have not conducted a camera calibration manually for the GR used in this study before, so it might be nonsense to show the results of automatic camera calibration. However, we believe that

the obtained DSM successfully show the trend of topographic changes at least in $10^{-1}$ order, as explained in the manuscript. Also, as suggested by the editor, we have provided the reprojection error values (page 6, lines 20–22).
* * *
Comments 8 (5-24) If you mention this, give an indication for how many locations and how they are distributed. Did you also include houses on the top-right (figures 2a-c)?

Authors: page 7, line 4 The text below has been removed.
*"This relationship was confirmed to be the same at several locations (not shown)"*
* * *
Comment 9 (6-2/4) How are these points distributed, with respect to each other and the GCPs. Include these in a figure.

Authors: See our response to Comment 6.
* * *
Comment 10 (6-9/19) You can calculate a limit of detection using these numbers and apply this in the figures.

Authors: The texts and figures below have been modified.

Page 7, lines 25–28: The text below has been added.
*"Then, the limit of detection (LoD) of elevation changes for each comparison pair was set to be the twice of the standard deviation of the systematic error between the DSMs, namely, 9.0 cm for the first two periods and 4.0 cm for the last two periods. The topographic changes less than this limit were neglected in the differential rasters and the volume calculation below"*

Page 7, lines 28: The text below has been removed.
*"topographic changes on the order of $10^{-1}$ m could be represented by these DSMs"*

Pages 31–32: The old Fig. 5 has been divided into two new figures, Fig. 6 and 7, applying the limits of detection. Also, see our response to Comment 13.
* * *
Comment 11 (6-13/14) Why didn't you use the lower of the two resolutions? Using the higher resolution can lead to local scale effects.

Authors: The SfM-derived DSM have been down-sampled to 1 m, which is equal to the resolution of September 2015

DSM, and applied to the comparison and the volumetric analysis. Figure 5 explains the effect of down-sampling to remove trivial relief of the original DSM. The texts and figures below have been modified.

Page 1, lines 13–14: the text below has been added.
*"and down-sampling of the SfM-derived DSM,"*

Page 7, lines 12–15: The text below has been added.
*"First, the resolution of the SfM-derived DSM was resampled to 1 m to avoid local scale effects that might be derived from trivial relief of the high-resolution surface model that was not represented in the DSMs with lower resolution (Fig. 5). The resolution and distribution of the pixels of the down-sampled DSM were set to the same as that of the September 2015 DSM for definite comparison of the two DSMs."*

Page 7, lines 20–22: The text below has been removed.
*"In the process of eliminating the systematic errors, the resolution of the SfM-derived DSM was reduced to 1/10 by computing the mean value of the original pixels with the Aggregate tool of the ArcGIS software because its original resolution was much higher than that of the other two DSMs."*

Page 7, line 25: The text below,
*"20.6 cm ± 1.8 cm"*
have been modified as below according to the results of the new DSM processing method.
*"20.5 cm ± 2.0 cm".*

Page 7, lines 31–32: The text below,
*"The resolution and pixel distribution of each differential raster were the same as those of the DSM with the higher resolution of each pair"*
has been modified as below.
*"The pixel sizes of them were both set to the 1 m for the convenience in volume calculation."*

Page 15, lines 4–5: The text below has been added.
*"and the adjusting of the resolution"*

Page 30: Figure 5 has been added.
* * *
Comment 12: For the applied systematic error correction you assume that the error is both linear and in the Z direction. Therefore it is important to: 1) show the distribution of the GCPs and control points (see earlier comments) and 2) show that there is no doming in the SfM DEM - this is a known problem see e.g. James and Robson (2014). James, M.

R. and Robson, S. (2014), Mitigating systematic error in topographic models derived from UAV and ground-based image networks. Earth Surf. Process. Landforms, 39: 1413–1420. doi:10.1002/esp.3609

Authors: In this study, we have used the following technics to prevent the dooming effect.

- Placed sufficient number of GCPs.
- Not used an automated camera gimbal, thus controlled vertical photos were not taken.
- The camera was flexibly held by a picavet (Inoue et al., 2014, Fig. 11), so off-nadir angles a little varied for each photo.
- Off-nadir angle was between 10–15 degrees backward.

In addition, a GNSS survey point which was not used as a GCP for SfM processing confirmed there is no dooming in the DSM. The texts below have been added. Also see our response to Comment 6.

Page 6, lines 11–13

*"The camera was flexibly held by a picavet (see Inoue et al., 2014) so that off-nadir angle of each photo was a little different among 10–15 degrees backward."*

Page 6, lines 22–29

*"The doming effect is a fundamental problem about DSM generation by SfM analysis associated with near-parallel image sets and inaccurate correction of radial lens distortion (James and Robson, 2014). The image acquisition method used in this study included some technics that mitigated the doming of the SfM-derived DSM: a high overlap rate of the images, precisely and widely placed GCPs and varying off-nadir angles. The error of each GCP between the GNSS measurement and the derived model was less than 5 cm, or the accuracy of the GNSS measurement, except for one located at the edge of the DSM. In addition, a check point was obtained by the same GNSS measurement showed an error of 0.56 cm against the SfM-derived DSM (Fig. 2). Although the result of the camera calibration was not quantitatively tested, these numbers suggest that the doming effect of the SfM-derived DSM was sufficiently small to compare with other LiDAR DSMs."*
* * *
Comment 13 (7-1) I think you should exclude inundated areas in the DEM of difference (either filter these areas out indicate them using a different color) - you could include them in a separate figure to indicate the water depth.

Authors: The text and figure have been modified as below.

Page 8, lines 1–6: The text below has been added.

*"In the inundated areas of the second DSM, apparent elevation changes to the water surface were represented by the first differential raster, rather than the true topographic changes (Fig. 8). The second differential raster showed the water depth there because all inundated areas were no longer existed in December 2015. Thus, the differential rasters*

were divided into two figures to separately show the topographic changes (Fig. 6) and elevation changes related to the water surface (Fig. 7). The inundated areas, which looked light brown due to the muddy water, were determined according to the aerial photograph of the second period (Fig. 3b)."

Page 31, 32: The Fig. 6 and 7 have been added.
* * *
Comment 14 (7-18) See earlier remark on limit of detection.

Authors: See our response to Comment 10. Also, the text has modified as below.

Page 9, lines 20–23: The text below has been removed because the similar information has already been provided in the method section (See our response to Comment 10).
"Areas where the differences between the two DSMs are less than 0.1 m after removal of the systematic error are transparent, showing the September 2015 orthophoto used as the background. Areas of the crevasse channel that were submerged in September 2015 show topographic changes related to the water surface, not the bottom of the crevasse channel, in the differential raster"
* * *
Comment 15 (8-8/9) Please support these comments with volume calculations.

Authors: Thank you for commenting. When the discussion paper was written, we had considered the flood sediments in the canal to be non-negligible compared to the entire volume of sediments. However, the volume calculation has revealed that is not the case. In the revised manuscript, more generic transportation of the materials was investigated (see our response to Comment 21), and the related texts have been revised as below.

Page 10, lines 21–23: The text below has been removed.
"In the case of the large canal on the east side of the research area, however, the removed sand had been transported elsewhere by December 2015."

Page 10, lines 23–26: The text below,
"Other than in those areas, the crevasse splay was left mostly undisturbed, so its features were deformed only by natural phenomena occurring during the 3 months after the flood."
has been modified as below.
"Although there were some points which experienced local aggradation or degradation due to similar restoration works as such, the crevasse splay was left mostly undisturbed, or had little modification so its features were deformed mainly by natural phenomena occurring during the 3 months after the flood, except for the reconstruction of the artificial levee and the land filling and leveling along the prefectural road near the breached levee."

Comment 16 (8-21/23): Unclear what you mean with 'important environmental component of the floodplain' and the relation with human changes. Do you mean that without human impact there would have been more vegetation in the floodplain and the crevasse splay topography might have been different (but how)?

Authors: In floodplains of natural rivers, the type and distribution of vegetation can be affected by emergence of crevasse splays by providing new deposits and creating new topography. In the2015 flood, however, new vegetation was lost through post-disaster restoration works, although that vegetation did exist around the sandy mound of crevasse splays in December 2015. They could potentially respond and record the nature of a crevasse splay and the adjacent floodplain if they were not removed by human.

Page 11, line 6–11 The text below,

*"..., thus, such new growth might be an important environmental component of the floodplain if there had been no human changes to the crevasse splay. Because each shrub is less than 1 m high, and they are distributed separately, they would be difficult to observe by LiDAR alone because of the limited resolution of the measured points. Usage of the UAV-SfM method to generate DSMs was therefore more suitable for documenting the relatively subtle changes in this case."*

has been modified as below.

*"This flood might provide a relationship between the topography of the crevasse splay and the location and the type of such new growth. However, such vegetation was removed during the restoration works"*
* * *
Comment 17 (8-31): Is the wind velocity sufficient to pick up the (fine) sediment?

Authors: In Joso City, the maximum wind velocity in a day sometimes exceeds 10 m s$^{-1}$ in winter, so it is possible to think the wind is responsible for the post-deposition deformation of the crevasse splay. However, the analysis of the wind process is not the scope of this study, and we believe that this topic should not be examined in detail

.

Page 11, lines 15–20 The text below,

*"This direction coincides with the direction of the seasonal wind in winter in Japan (Table 3). Available wind data from after the flood show that on 63 of 100 days the dominant wind direction was between northerly and westerly, which could account for the low gradient of the splay edges. The dry wintertime climate of Japan might also help the sand to be easily moved by the wind"*

has been modified as below.

*"The seasonal wind in winter in Japan, the direction of which coincides with the sand and occasionally exceeds 10 m s$^{-1}$ in speed is possibly responsible for the change combined with the dry wintertime climate of Japan"*

 The text below has been removed.

"The climate data in Joso City are freely available from Japan Meteorological Agency (http://www.jma.go.jp/jma/menu/menureport.html)."

 The original Table 3 has been removed.
* * *
Comment 18 (9-7/9) What do you exactly mean with 'simpler topography'? Would it have been possible if the topography was more 'complex' at the later time?

Authors: The texts below have been removed because the comparison method have been changed in the revised manuscript and these statements became inappropriate.

 The text below has been removed.
*"The resolution of the LiDAR-derived DSM is much lower than the resolution of that produced by the UAV-SfM method, but it was possible to compare the topography at finer scale than the LiDAR data by fitting the pixels of the differential raster to the UAV-derived DSM, because actual topography at the earlier time was simpler than that at the later time. In particular, where the topographic relief is small and vegetation is sparse, as in the study area, even relatively low resolution topographic data is adequate for describing and archiving the effects of a disaster on the topography through comparisons with high-resolution data obtained after the event."*

 The text below has been removed.
*"The relatively low resolution of the LiDAR-derived DSM limited detailed quantification of the topography, but subtle changes on a horizontal scale smaller than the pixel size of the LiDAR-derived DSM were detectable in the comparison with a DSM acquired later by the UAV-SfM method because the topography in the earlier DSM was relatively simple."*
* * *
Comment 19 (9-21/22) Note that it may also be valuable to research historical events using archival photogrammetry, e.g. Bakker and Lane (2016). Bakker, M., and Lane, S. N. (2016), Archival photogrammetric analysis of river–floodplain systems using Structure from Motion (SfM) methods. Earth Surf. Process. Landforms, doi: 10.1002/esp.4085.

Authors:  The text below has been added.
*"Another approach to quantifying morphological changes is archival photogrammetry using SfM methods with past aerial photographs (Bakker and Lane, 2016). Although data sources may be limited in number and resolution, this approach can reconstructing historical events and thus extend the timeline of topography to the period before LiDAR data have been available."*

Comment 20 (9-31/34) This is a very important point and I think you have a good data set to include these calculations!

Authors: The text below,

Page 14, lines 17–19

"Applications suggested by the results of this study include volume estimation of flood deposits that need to be removed and, conversely, the amount of materials necessary for the construction of embankments or for land filling. Evaluation of these possibilities, however, exceeds the scope of this paper."

has been modified as below and moved to Page 13, lines 7–11

"Nevertheless, quantitative information derived from a time-series of topographic data fairly represents 3D architecture of a crevasse splay and may enable us, in disastrous situations, to consider the amount of deposits that need to be removed and, conversely, the amount of materials necessary for the construction of embankments or for land filling, In the case of this study, volumetric contribution of sand lobes and land filling to the topography might be suit for these applications."

Comment 21 Most important remark concerning the RESULTS AND DISCUSSION is that a volumetric analysis would be very valuable (for the area as a whole and / or for certain regions), indicating the net amount of sediment that came from the river channel, the amount of erosion / sedimentation on the levee/floodplain, the amount of sediment that was redistributed / imported during post-flood works.

Authors: According to the suggestion of the reviewer, we have conducted the volumetric analysis of the flood using the DSMs. The procedure, results, and insights obtained by the analysis have been described in a newly added section, "*5.3 Volumetric evaluations of the topographic changes*" as below. Also, additional tables and figures have been made to explain the procedure and the results. Other parts of the manuscript have been modified to include the volumetric analysis.

Page 11, line 21–page13, line 11: The section 5.3 have been added.

Pages 23, 24, 33, 34: Two tables (Tables 3 and 4) and two figures (Fig. 8 and 10) have been added ().

Page 1, lines 16–18: The text below has been added.

"If excluding changes other than topography, comparison of the DSMs showed that more than twice as large volume was eroded than deposited 300–500 m around the breached artificial levee where the topography was significantly affected "

Page 2, lines 2–4: The text below has been added.

"volume calculation was conducted using these DSMs to investigate the balance between deposition and erosion processes acted in the research area during the flood and the volumetric extent of the post-flood restoration works against the breached artificial levee and intensively eroded areas"

Page 14, line 6: The text below has been added.

"and volumetric estimation of the topographic changes".

Page 14, line 26: The text below has been added.

"However, these objects might have considerable effects in case of volumetric analysis."

Page 15, lines 7–11: The text below has been added.

"The topography of the crevasse splay was characterized by the intensive erosion near the breached levee and deposition of the flood deposits, especially lobe-shaped sand mounds occupying 33.5% of the all aggradation, surrounding the erosional areas. Volumetric analysis indicated that the area of the degradation pixels was about three times narrower than the aggradation pixels but the estimated volume loss was more than twice as large as volume gain, resulting in the volume loss of 43 239 m³ in all by the event which was suggested to be quite erosive in the research area."

Page 15, lines 14–17: The text below has been added.

"The true erosional depths in the inundated areas could be estimated using the DSM in the third period, resulting in improving by 8.1% of the estimation of the total volume loss. The reconstruction of the breached levee and the leveling of intensively eroded ground near it were major causes of the topographic changes after the occurrence of the flood and the value countervailed 25.6 % of the eroded volume by the flood."

Page 15, lines 21–26: The text below has been added.

"However, the volume estimation in this study have some issues including (i) effects other than the topography such as buildings and agricultural crops, (ii) lack of topographic data showing the entire part of the crevasse splay: partly the erosion depths could not be estimated due to the restoration works, and (iii) if only a single process worked at a point, namely, erosion in the aggradational pixels and deposition in the degradational pixels were neglected. Use of topographic models in which buildings and vegetation are removed, more frequent measurement of the topography, which may easily be achieved by the UAV-SfM protocol, and the combination with field observation can improve the estimation."

Page 23: Table 3 has been added.

Page 24: Table 4 has been added.

Page 33: Figure 8 has been added.

: Figure 10 has been added.
* * *
**TECHNICAL CORRECTIONS**

TC 1: Title - You mention 'multitemporal', but in this case (3 measurements) 'repeated' is perhaps more appropriate.

Authors: The title of our manuscript have been changed as "Application of UAV-SfM photogrammetry and aerial LiDAR to a disastrous flood: repeated topographic measurement of a newly formed crevasse splay of the Kinu River, central Japan". In other parts of the manuscript, "multitemporal" have been replaced by "repeated".

TC 2: 1-14 'by subtraction' can be removed.

Authors: 'by subtraction' has been removed.

TC 3: 1-16 'carried out by people' can be removed.

Authors: 'carried out by people' has been removed.

TC 4: 1-17 'with different resolutions and acquisition periods': I would say different spatial and temporal resolutions. (Acquisition period can be interpreted as the duration of acquisition, e.g. the flight time of the UAV).

Authors: The text was removed through the reorganization.

TC 5: 1-18 'sudden' can be removed.

Authors: 'sudden' has been removed.

TC 6: 2-4 It is questionable if you should include a reference to this unpublished paper. Both here and further in the manuscript this reference is not required / of added value. I would advise not to include it.

Authors: This unpublished paper have been removed from the reference list. Instead, a published paper, Matsumoto et al. (2016), that have documented the characteristics of the flood deposits of this event have been cited (page 13, line 5).

TC 7: 2-4 'Thus' can be removed (there is no direct link with the previous sentence).

Authors: 'Thus' has been removed.

TC 8: 3-3 Refer to the figure here.

Authors: Figure 1 has been referred.

TC 9: 4-22 Brackets can be removed here and later on when mentioning points/m2.

Authors: They have been removed.

TC 10: 4-28 Include that this was converted to raster (similar to the pre-flood lidar data).

Authors: That has been included.

TC 11: 5-8/10 Only mention the usable photos.

Authors: Page 8, lines 8–11: The text below,
"1433 photos were taken at 2-s intervals. However, these included many unusable photos that were taken when the UAV was not flying along the planned path, for example, while it was ascending or descending (5 m s–1 and 2.5 m s–1). Finally, of the 1433 photos, 597 photos with an 80% overlap both across and along the UAV paths were analyzed with the SfM technique to generate a DSM"
has been modified as below.
"597 photos were taken at 2-seconds intervals with an 80% overlap both across and along the UAV paths"

TC 12: 6-17/27 This is a (specific) description of the study area - not results.

Authors: This part have been moved to the third section and modified as below.

Page 3, lines 24–31
"The research area covered by the DSMs was on the southern part of an alluvial ridge along the Kinu River, the width of which was about 1.5 km (Fig. 1). Within the research area, the elevation of the ridge was high along the channel, especially near the breached levee, although the ground level there might have been raised for construction. The artificial levee was about 2 m higher than the top of the alluvial ridge. The alluvial ridge was divided by a shallow valley oriented north–south along the eastern margin of the study area, which Sadakata (1971) suggested was a past crevasse channel. In the present, an agricultural canal ran along the valley, and a network of smaller canals covered the research area. The part of the alluvial ridge in the study area was mainly used for cultivating agricultural crops; most of the area was not covered by pavement or buildings except close to the levee."

TC 13: 7-1 This part of the method.

Authors: It has been moved to the method section and modified as below.

page 5, lines 24–26
*"When the LiDAR data in September 2015 was acquired, however, some areas (mostly erosional zones) were inundated by the flood water, which means that elevation data cloud not be obtained there. Thus, the elevation of the inundated areas was interpolated from the elevations along the margins of those areas on the DSM."*

TC 14: Table 1: This table can be removed (little information and no additional information than in the text).

Authors: It has been removed.

TC 15: Table 3: What is the (estimated) velocity required to transport sediment? Mean velocity
is probably not a suitable measure, perhaps the 90th percentile(?).

Authors: This table has been removed because the wind process is not the point of our study (see our response to Comment 17).

TC 16: Figure 1: Include 'Japan' in the left top figure.

Authors: 'Japan' has been included.
* * *
**Referee 2**

**GENERAL COMMENTS**

This study is very interesting, because it shows the detail of topographic changes that occurred during a levee breach / crevasse splay. And it shows the usefulness of UAV-SfM method. I judge that this study is worth for publication in this journal.

**TECHNICAL CORRECTIONS**

P4 20-21 Kanto Regional Development Bureau, Land, Infrastructure and Transportation Ministry of Japan
P10 29-30 Kanto Regional Development Bureau, Ministry of Land, Infrastructure, Transport and Tourism
Right indication Kanto Regional Development Bureau, Ministry of Land, Infrastructure, Transport and Tourism

Authors: The correction in the text and the reference list has been done.
* * *
Authors: Other than suggested revision above, there are some modifications done by the authors for more explanation,

or consistency with the suggested revision. List of the important changes are shown below.

1.

page 1, lines 14–20: The abstract has been reorganized to apply the suggestions by the reviewer and the editor. The text,

*"After elimination of systematic errors among the DSMs, differential DSMs were produced by subtraction and topographic changes on the order of 10–1 m were detected. These changes were found to be consistent with previously reported ground survey data. The detected changes included not only topographic changes but also growth of vegetation, vanishing of floodwaters, and restoration and repair works carried out by people."*
*has been modified as below.*
*"After elimination of systematic errors among the DSMs and down-sampling of the SfM-derived DSM, elevation changes on the order of $10^{-1}$ m including not only topography but also growth of vegetation, vanishing of flood waters, and restoration and repair works were detected. If excluding changes other than topography, comparison of the DSMs showed that more than twice as large volume was eroded than deposited 300–500 m around the breached artificial levee where the topography was significantly affected."*

2.

page 1, lines 22–23: The text below has been removed.
*"Moreover, they have the great advantage that they can be used to archive such changes that occur in residential areas and urban areas where their preservation potential is low."*

3.

page 1, line 24–page 2, line 1: The text,
*"…crevasse splays (sand splays), that is, discrete lobes or finger-shaped mounds of sandy sediments, and…"*
has been modified as below for the consistency in the terminology.
*"…crevasse splays, which are mainly composed of sand lobes, that is, discrete lobe- or finger-shaped mounds of sandy sediments, and…"*

4.

page 2, lines 32: The text below has been added.
*"characterized by the formation of crevasse splay"*

5.

page 3, lines 4–5: The text below has been removed.
*"the comparability of topographic data obtained by different acquisition methods and with different resolutions and the use of reference points was examined."*

6.

page 4, line 21: The text below has been added.

*"occurred at 10 September"*

7.

page 6, lines 17–18: The text below has been added.

*"so that local relief and objects would not affect the elevation of the pixels around the GCPs"*

8.

page 7, lines 15–16: The text,

*"Before the differential rasters could be calculated,"*

Has been modified as below.

*"Before the calculation of the differential rasters"*

9.

page 8, line 8–page 9, line 8: The original section "5.1 Overview of the topography of the alluvial ridge and crevasse splay on each date" has been removed and the paragraphs within have been moved to other sections. The first, second, and third paragraphs have been moved to section 2 and new 5.1 and new 5.2, respectively.

10.

page 4, lines 11: The text below has been added.

*"especially around the edge of the splay deposits"*

11.

page 13, lines 13–17: The text,

*"In this study, the systematic error between DSMs was assumed to be merely the average of the elevation differences at several selected points. In fact, almost all parts of the road, which were assumed to be stable throughout, were shown to experience small elevation changes (<10 cm) in both comparison pairs (Fig. 5). This result suggests that even the simple elimination method used in this study can effectively detect differences on the order of 10–1 m in topographic data acquired by different methods and at different times."*

has been moved to the Conclusions section and modified as below.

page 13, line 28–page 15, line 3

*"The systematic error between DSMs was assumed to be merely the average of the elevation differences at several selected points. In fact, almost all parts of the road, which were assumed to be stable throughout, were shown to experience small elevation changes (less than 10 cm) in both two comparison pairs, January 2007 versus September 2015 and September 2015 and December 2015. However, very high resolution of the UAV-SfM derived DSM might be inappropriate to compare with the LiDAR derived DSMs with relatively low resolution, due to the trivial relief of the surface model derived from wheel tracks, plants, debris and so on. Down-sampling of the former DSM to 1 m, which was equal to that of LiDAR derived DSM, removed these local fluctuation of the topography, enabling more easy comparison of the surface models acquired by LiDAR."*

12.

page 14, line 3: The text below has been added.

*"combined with SfM analysis"*

page 14, lines 14–17: The text below has been moved to the head of this section (page 13, lines 24–27).

*"In the context of disaster management, high-resolution photos and videos taken by UAVs can help in the development of strategies to take in the event of emergencies (Ezequiel et al., 2014; Erdelj and Natalizio, 2016). Another UAV 'eye', namely, topographic data obtained by SfM photogrammetry, would also be informative, particularly if acquired immediately after the occurrence of a disaster."*

14.

Page 34: In Figure 9, the photos taken in June and August 2016 in the original (a) and (b) have been removed because these photos are not mentioned in the text. The caption has also been modified accordingly.

[revised manuscript text omitted]

---

## Author Response (AR2)

**Response to the Editor's comments**

Authors: Thank you for your interests about our paper and valuable comments to improve it. We would like to respond to the comments in point-per-point manner. According to the comments, we have made some modifications to the manuscript. Please check our response to the comments and marked-up manuscript below.
Note that this manuscript was checked by two native English speakers and wording and phrasing were significantly changed from the original.

**Specific comments**
* * *
Comment 1

Page 1 Lines 15-17 Please make it concise.

Authors

page 1, lines 15–17: The text below,

*"If excluding changes other than topography, comparison of the DSMs showed that more than twice as large volume was eroded than deposited 300–500 m around the breached artificial levee where the topography was significantly affected."*

has been modified to,

*"Comparison of the DSMs showed that the volume eroded by the flood was more than twice the deposited volume in the area within 300–500 m of the breached artificial levee, where the topography was significantly affected."*
* * *
Comment 2

Page 1 Lines 22-25, 25-29 Redundant.

Authors

Page 1, lines 22–29: The texts below and relavant references have been removed.

*"Floodplains have long been desirable locations for human habitation owing to their fertile soils and abundant water availability, and many of the world's great cities have been constructed on and have drastically modified the natural environment of floodplains, with the result that flooding in these areas, when it occurs, can be extremely hazardous (Wohl, 2014). When the levee of a river with a sandy bed collapses, crevasse splays, which are mainly composed of sand lobes, that is, discrete lobe- or finger-shaped mounds of sandy sediments, and crevasse channels (ephemeral tributaries through which water and sediments are transported onto the floodplain) form, resulting in abrupt, severe topographic changes on the floodplain (Allen, 1965; O'Brien and Wells, 1986; Bristow et al., 1999; Florsheim and Mount, 2002; Day et al., 2016)"*
* * *
Comment 3

Page 2 Lines 11-12 Just "(Gomez and Purdie, 2016)" would be enough.

Authors

Page 2, lines 12–13: Modified to "*(Gomez and Purdie, 2016)*".
* * *
Comment 4

Page 2 Line 14 Insert "aerial" before "LiDAR"

Authors

Page 2, line 15: The text has been modified according to the suggestion.
* * *
Comment 5

Page 2 Line 16 Rephrase as "The UAV-based SfM photogrammetric method (UAV-SfM)"

Authors

Page 2, line 17: The text has been modified according to the suggestion.
* * *
Comment 6

Page 2 Line 18 "gravelly" >> "gravel-bed"

Authors

Page 2, line 19: The text has been modified according to the suggestion.
* * *
Comment 7

Page 2 Line 18 It seems that the sentences after the second "Although" is not specifically related to the "high-resolution topographic data" but to more general issue in the floodplain studies. Can it be a new paragraph?

Authors

Page 2, lines 20–26: The text "Although several studies~" has been moved to a new paragraph.
* * *
Comment 8

Page 2 Lines 25- The sentences can be rephrased as:

"In this study, these difficulties were challenged using high-resolution topographic datasets for the case of a disastrous flood along the Kinu River, central Japan in 2015, which affected an inhabited and cultivated floodplain characterized by the formation of crevasse splay. Three digital surface models (DSMs) of the research area were generated by aerial LiDAR and UAV-SfM: before, 3 days after, and 3 months after the flood.

The features and post-formation modification of thetopography caused by a levee breach were then quantitatively documented from the perspective of both natural and artificial changes. In addition, volume calculation was conducted using these DSMs to investigate the balance between deposition and erosion processes during the flood, and the volumetric extent of the post-flood restoration works in the breached artificial levee and intensively eroded areas."

Authors

Page 2, line 27–Page 3, line 2: The text has been modified according to the suggestion and the native check as below.

"In this study, we used high-resolution topographic data sets to deal with these difficulties and studied topographic changes caused by a disastrous flood that occurred along the Kinu River, central Japan, in 2015. In the study area, this flood caused a crevasse splay to form on an inhabited and cultivated floodplain. Three digital surface models (DSMs) of the research area were generated by aerial LiDAR and UAV-SfM: before, 3 days after, and 3 months after the flood. The features and post-flood modification of the topography caused by a levee breach were then quantitatively documented from the perspective of both natural and artificial changes. In addition, volume calculations were conducted using these DSMs to investigate the balance between deposition and erosion processes during the flood, and the volumetric extent of the post-flood restoration works in the breached artificial levee and intensively eroded areas."

Comment 9

Page 3 see the attached file.

Authors

Section 2 Study area: The section has been modified according to the suggestion.

Page 3, line 4: The texts "which occupies the center of the floodplain. The floodplain itself is 4–8 km wide, and it is bordered on both the east and west by fluvial terraces (Fig. 1)" have been removed.

Comment 10

Page 4 Lines 17-20 Better to split this long sentence.

Authors

page 4, lines 17–20: The text,

"It took 10 days of pumping to remove the flood water from the levee-protected floodplain; there were two deaths, 44

*injured, and 6000 evacuees of the 65,000 inhabitants of Joso City, and the flood damaged, destroyed, or inundated*

*5000 buildings, in addition to causing severe interruptions of public utilities and the transportation system (Joso City,*

*2016)"*

has been modified to below, ommiting redandunt information.

*"The flood caused two deaths, injured 44, and forced the evacuation of 6000 of the 65 000 inhabitants of Joso City, and*

*it damaged, destroyed, or inundated 5000 buildings (Joso City, 2016)."*
* * *
Comment 11

Page 4 Line 22-23 "159.2 billion yens in Ibaraki Prefecture" could be unnecessary. "yens" >> "JPY"

Authors

Page 4, line 23: The text has been modified according to the suggestion.
* * *
Comment 12

Page 4 Lines 24-25 Rephrase and shorten the sentence below, or just remove it because the reference is already cited on Line 17.

"For further information of 2015 flood of the Kinu River, see Nagumo et al. (2016) which have provided a detailed report of damages and social effects in the flooded areas."

Authors

Page 4, lines 24–26: The suggested text has been removed.
* * *
Comment 13

Page 5 Line 3 "fairly accurate" >> "fairly clean to represent the bare land"

Authors

Page 5, line 8: The text has been modified according to the suggestion and the native check.
* * *
Comment 14

Page 5 Line 15 What kind of interpolation method did the authors use?

Authors

Page 5, lines 18–25: In general, elevation of water surface and topography under water cannot be obtained by LiDAR. However, it is found that the measurement just after the flood was almost succcessful in the inundated areas probably

because of the muddy water. The calculation results were not affected at all by this notice. We have modified the manuscript as below.

*"When the LiDAR data in September 2015 were acquired, some areas (mostly erosional zones) were inundated by floodwaters. However, laser scanning data could still be obtained with the same density in such inundated areas, probably because the muddy water was able to reflect the laser beams. The elevation of the inundated areas was interpolated to a grid in the same way as the non-inundated areas. It should be noted that whether or not the laser beams reflected strictly from the water surface was not confirmed. In this study, however, they were regarded as indicating the elevation of the water surface because the elevation of continuous water bodies showed little fluctuation and matched that of the ground at the edge of the water."*
* * *
Comment 15

Page 5 Line 16 Is this RTK-GNSS?

Authors

Page 5, lines 26–29: A kinematic GNSS measurement was used during the flight operation. The text has been modified to include more information about the positioning procedure as below.

*"During the LiDAR measurement, the position of the aircraft was determined by kinematic GNSS equipment, and the inclination of the aircraft (roll, pitch, and yaw) was logged by an inertial measurement unit. The on-board position information was improved by post-processing using ground reference stations maintained by the Geospatial Authority of Japan, established by a static GNSS survey."*
* * *
Comment 16

Page 5 Line 17 "internal" >> "inertial"

Authors

Page 5, line 27: The text has been modified according to the suggestion.
* * *
Comment 17

Page 5 Line 20 What is "APM"?

Authors

Page 6, lines 2–3: APM means ArduPilot Mega, a UAV flight controller by 3DR. The text has been modified as below.

*"a UAV (DJI F550 six-rotor multicopter equipped with an ArduPilot Mega 2.6, UAV flight controller by 3DR)"*
* * *
Comment 18

Page 5 Line 25 "focal length, " >> "focal length: "

Authors

Page 6, line 7: The text has been modified according to the suggestion.
* * *
Comment 19

Page 6 Line 3 "west direction, and 1.58 cm" >> "west direction, 1.58 cm"

"0.455" >> "0.46"

Authors

Page 6, line 18: The text has been modified according to the suggestion.
* * *
Comment 20

Page 6 Line 8 "technics" >> "techniques"

Authors

Page 6, line 22: The text has been modified according to the suggestion.
* * *
Comment 21

Page 6 Line 15 "gradient" >> "slope gradient"

Authors

Page 6, line 31: The term has been modified as "pixel gradient" as it seems to be common. The same term is used in the other parts of the manuscript.
* * *
Comment 22

Page 6 Line 19 "set to the same" >> "set to be the same"

Authors

Page 7, line 13: The text has been modified according to the suggestion.
* * *
Comment 23

Page 7 Line 14 remove "were" before "no longer"

Authors
Page 7, line 31: The text has been modified according to the suggestion.
* * *
Comment 24

Page 7 Line 29 "shows" >> "show"

Authors
Page 8, line 14: The text has been modified according to the suggestion.
* * *
Comment 25

Page 8 Lines 27-31 The sentence below is too long and hard to read.

"Althoughthere were some points which experienced local aggradation or degradation due to similar restoration works as such, thecrevasse splay was left mostly undisturbed, or had little modification so its features were deformed mainly by natural phenomena occurring during the 3 months after the flood, except for the reconstruction of the artificial levee and the landfilling and leveling along the prefectural road near the breached levee."

Authors
Page 9, line 14–17: The text has been modified as below.

"*Although similar restoration works caused local aggradation or degradation at some points by December 2015, the crevasse splay was left mostly undisturbed, or was modified very little. Thus, during the 3 months after the flood, its features were changed mainly by natural phenomena, except for the reconstruction of the artificial levee and the land filling and leveling along the prefectural road near the breached levee.*"
* * *
Comment 26

Page 12 Line 4 "resolution" >> "horizontal resolution"

Authors
Page 12, line 29: The text has been modified according to the suggestion.
* * *
Comment 27

Page 13 Lines 1-4 The sentence below is too long and hard to follow. I would recommend to split them. Also, the third point is unclear.

"However, the volume estimation in this study have some issues including (i) effects other than the topography such as buildingsand agricultural crops, (ii) lack of topographic data showing the entire part of the crevasse splay: partly the erosion depthscould not be estimated due to the restoration works, and (iii) ifonly a single process worked at a point, namely, erosion in the aggradational pixels and deposition in the degradational pixels were neglected."

Cooment 28

Page 13 Lines 4-6 For the improvement of volume estimation, the better filtering of the data makes sense, but it is unclear why more frequent measurements contribute. Please explain.

Authors

Page 13, line 28–page 14, line 2: The text has been divided into each point. The third point means that the timing is important for measurements of rapidly changing topography such as natural disasters. In this study, the complete topographic data of crevasse splay could not be obtained because the restoration works began before the measurement. Thus, measurements in appropriate periods are necessary to purchase the changing topography. The texts have been modified as below.

[revised manuscript text omitted]

---

## Author Response (AR3)

**Response to the Editor's comments**

Authors: Thank you for commenting to our paper. We would like to respond to the comments in point-per-point manner. According to the comments, we have made some modifications to the manuscript. Please check our response to the comments and marked-up manuscript below.
* * *
1) This is not mandatory, but if applicable, I would recommend describing thanks to the reviewers, as well as to the native English speakers who supported revising the manuscript in Acknowledgements.

Authors

page 14, lines 5–8: Acknowledgements section was rewritten as below.

*"We gratefully thank Hiroshi Kobayashi of Aero Asahi Corporation for providing the processed DSM derived from the post-flood LiDAR data and the orthophotos acquired in September 2015. This study was financially supported by JSPS KAKENHI Grant Number JP26282078. We would like to thank the one anonymous reviewer and Yuichi S. Hayakawa for their constructive comments, which helped to improve the manuscript. Language editing was provided by Susan Duhon and Bob Wathen of ELSS, Inc."*
* * *
2) Page 5 Lines 18-25: "The elevation of the inundated areas was interpolated to a grid in the same way as the non-inundated areas." can be modified to:

"The elevation of the inundated areas was interpolated to a grid in the same way as the non-inundated areas (TIN method)."

Authors

Page 5, lines 3–4: The text was changed according to the suggestion.
* * *
3) Page 6 Line 31: "The term has been modified as 'pixel gradient' as it seems to be common. The same term is used in the other parts of the manuscript."

I am unfamiliar with the term "pixel gradient" as a term representing the slope of topography using DEM (either DSM or DTM). I think "pixel gradient" may fit other types of raster data (RGB etc.), while "slope gradient" directly represents the topographic slope. Also, please clarify what method was used to calculate the gradient (conventional eight cells method?).

Authors

Page 6, lines 11–15, Page 25: "Pixel gradient" was replaced by the suggested term, and the method by which the gradient was calculated was described as below.

*"The slope gradients of each rasters were calculated using the Slope tool of ArcGIS with the planar method."*
* * *
4) Page 13 Line 10: "that assumption is not valid." can be modified to:

"that assumption is not validated."

Authors

Page 13, line 10: The text was changed according to the suggestion.

[revised manuscript text omitted]

(a) Dec 2015: Original DSM          (b) Dec 2015: Down-sampled DSM

(c) Dec 2015: Orthophoto         (d) Sep 2015: DSM

[Figure]

100 m

Figure 5: Enlarged views of Fig. 3 images for comparison of the DSMs with different resolutions. (a) Original DSM for December 2015 (resolution: 3.84 cm). (b) Down-sampled DSM for December 2015 (resolution: 1 m). (c) Orthophoto taken in December 2015. (d) DSM for September 2015 (resolution: 1 m). Note that the trivial relief in (a) is largely removed in the down-sampled DSM in (b).

[Figure]

**Figure 6: Differential rasters indicating the elevation differences between two successive snapshots after the removal of the systematic error and the adjustment of resolution. (a) The January 2007 DSM subtracted from the September 2015 DSM. Note that erosion near the breached levee was more than 1 m. (b) The September 2015 DSM subtracted from the December 2015 DSM. Note that in DSM areas inundated in September 2015 are masked (see Fig. 8 for the elevation changes in these areas).**

[Figure]

**Figure 7: Differential rasters indicating the elevation differences between two successive snapshots after the removal of the systematic error the adjustment of resolution in the areas inundated in September 2015. (a) The January 2007 DSM subtracted from the September 2015 DSM. Note that erosion near the breached levee was more than 1 m. (b) The September 2015 DSM subtracted from the December 2015 DSM. The elevation changes within the dashed line do not represent water depths because artificial land filling has been done.**

[Figure]

January 2007

Original ground surface

September 2015

Aggradation

Degradation

Sand lobe

Inundated

LoD

Erosional depth

LoD

Detected change in elevation (Fig. 7a)

December 2015

Sand lobe

Land filling

LoD

LoD

Actual erosional depth

Inundated depth (Fig. 7b)

Estimated inundated depth

**Figure 8: Schematic description of the topographic changes caused by the flood and the later disappearance of inundation water and land filling. Note that elevation changes smaller than the LoD were neglected in the differential rasters and the volume calculations.**

[Figure]

[Figure]

[Figure]

[Figure]

**Figure 9: Photos of the study area. (a) Crevasse channel (breach scouring) near the levee. Note the building still standing after the flood in the center of the photo. (b) Sand splay in the southern part of the study area. (c) Damaged prefectural road beside the breached levee. (d) Overturned and transported house. Note the sandy mound in the downstream direction from the house. See Fig. 3c for the photo locations. Photos were taken by A. Izumida and T. Sugai.**

[Figure]

**Figure 10: The calculation range used for the volumetric evaluation and the topography types used in the calculation. The areas that were inundated but not leveled in December 2015 (green) contain some aggradational pixels; these pixels were not used in the estimation of water volume, but they were included in the volume gain.**